# Feedforward attentional selection in sensory cortex

**Jacob A. Westerberg** [1,2,3,4] ✉, **Jeffrey D. Schall** [5,6,7,8],
**Geoffrey F. Woodman**[1,2,3] **& Alexander Maier** [1,2,3]

Salient objects grab attention because they stand out from their surroundings. Whether this phenomenon is accomplished by bottom-up sensory processing or requires top-down guidance is debated. We tested these alternative hypotheses by measuring how early and in which cortical layer(s) neural spiking distinguished a target from a distractor. We measured synaptic and spiking activity across cortical columns in mid-level area V4 of male macaque monkeys performing visual search for a color singleton. A neural signature of attentional capture was observed in the earliest response in the input layer 4. The magnitude of this response predicted response time and accuracy. Errant behavior followed errant selection. Because this response preceded top-down influences and arose in the cortical layer not targeted by top-down connections, these findings demonstrate that feedforward activation of sensory cortex can underlie attentional priority.

We constantly filter through our complex environment to extract information pertinent to our goals. In this effort, some objects seem to grab our attention when they differ from their surroundings. However, the mechanisms supporting this attentional prioritization through salience-based capture ("pop-out") remain elusive[1–3]. The behavioral and phenomenal consequences associated with pop-out are frequently described as "exogenous attention" or "stimulus-driven attention"[4], suggesting that feedforward sensory processes take a preeminent role. However, this has never been demonstrated directly. As a consequence, attentional prioritization through salience-based capture has been theorized to arrive automatically and feedforward[3] or via cognitive mediation[1]. An intermediate hypothesis proposes that an automatic priority signal is generated in response to attention-grabbing objects, which is biased to promote behaviorally useful objects[2].

With respect to the underlying neurobiology, each account has dissociable putative neural mechanisms. Under the feedforward account, it is hypothesized that the stimulus-driven neuronal response to an attention-capturing object defines the selection process. With the understanding that visual information propagates through a hierarchy of brain areas and their respective microcircuitry (e.g., canonical laminar activation patterns within and across brain areas)[5–7], feedforward attentional capture should be observed in earlier brain areas and within feedforward-recipient cortical layers before later brain areas and feedback-recipient cortical layers. This same spatiotemporal framework can be inverted to serve the alternative hypothesis. That is, in the feedback hypothesis, it is entirely plausible that the selection process descends the visual processing hierarchy following the feedforward cascade of neuronal activation. However, it is important to note that these hypotheses need not be mutually exclusive. Neuronally, the intermediate hypothesis relies on feedforward attentional capture, which is modulated by feedback processes, an interactive process with established evidence in the early visual system in other perceptual and cognitive tasks[8].

We tested the predictions of these competing theoretical views by recording across the layers of primate area V4 while macaque monkeys searched for oddball targets in arrays of objects. Area V4 is ideal for testing predictions of the competing accounts of attentional capture

---

[1]Department of Psychology, Vanderbilt University, Nashville, TN 37240, USA. [2]Vanderbilt Brain Institute, Vanderbilt University, Nashville, TN 37240, USA. [3]Vanderbilt Vision Research Center, Vanderbilt University, Nashville, TN 37240, USA. [4]Department of Vision and Cognition, Netherlands Institute for Neuroscience, Royal Netherlands Academy of Arts and Sciences, 1105 BA Amsterdam, The Netherlands. [5]Centre for Vision Research, York University, Toronto, ON M3J 1P3, Canada. [6]Vision: Science to Applications Program, York University, Toronto, ON M3J 1P3, Canada. [7]Department of Biology, York University, Toronto, ON M3J 1P3, Canada. [8]Department of Psychology, York University, Toronto, ON M3J 1P3, Canada. ✉e-mail: j.westerberg@nin.knaw.nl

as it receives from, and projects to, both earlier visual cortical areas and higher-order cortex implicated in cognitive control[9,10] while showing robust attentional modulation[11] during visual search[12,13].

We found neural signatures underlying attentional capture occur within the earliest synaptic and spiking activation of the feedforward-recipient layers of V4 that comprise the initial spatio-temporal volley of sensory responses. This finding is incompatible with hypotheses involving extensive feedback from higher areas. Instead, these data suggest that "exogenous" or "stimulus-driven" attentional priority largely arises from bottom-up processes in the early sensory cortex.

## Results

The feedforward account of attentional capture posits that attention-grabbing objects automatically engender capture[2,3]. For this account to hold, feedforward sensory activation elicited by these objects should predict behavioral measures of attentional capture, such as reaction time. This finding would suggest the priority[14] of attention-grabbing objects is computed rapidly and in the sensory cortex[15]. Priority indexes the utility (as opposed to a sensory feature) of a stimulus. Salience (here, a sensory feature) refers to the physical distinctiveness of a stimulus relative to its context. In pop-out, salience and priority are tightly coupled, yet these two attributes are experimentally distinguishable since a non-salient stimulus can sometimes be (erroneously) chosen as having the highest utility. However, feedforward sensory activation has not been shown to be tightly coupled to behavioral accuracy and reaction time in attention-demanding search tasks.

In contrast, the feedback account of attentional capture puts forward that stimulus features are prioritized as a function of a process distinct from the feedforward sensory response. In the extreme case, this view predicts modulation of cortical activity in the absence of visual stimulation. The latter phenomenon might manifest, for example, as persistent changes to ongoing activity, or during intertrial periods[11,16,17]. Top-down attentional modulations of neural activity can manifest when spatial selective attention is deployed[18–39], but whether this feedback-driven mechanism of attentional modulation is also instantiated for pop-out selection is an open question. Finally, on the intermediate account, we expect both mechanisms to emerge.

### Pop-out visual search behavior and neural responses

Two monkeys performed a color-based pop-out visual search response time (RT) contingent task (Fig. 1a). Monkeys would fixate a fixation point on a visual display and following a variable delay, be presented with an array of 6 equally spaced red or green circles at a fixed eccentricity around the fixation point. One circle was of a saliently different color than the rest (e.g., 1 red oddball target among 5 green distractors or vice versa). Monkeys, as quickly as possible, made an eye movement to the oddball stimulus (i.e., the RT). Trials were organized into blocks where monkeys would search for either a red or green circle for 5 to 15 consecutive trials before the target/distractor identities swapped. For example, monkeys would search for red among green for 7 trials then green among red for 12 trials, and so forth. The length of a given block was unpredictable. If the monkeys made a correct eye movement, they were rewarded.

Monkeys performed this task well above chance (Session accuracy averages: monkey Ca, 88%; monkey He, 81%; chance, 16.67%) with RTs comparable to previous reports of monkeys performing the same task[40–42] (Session RT averages: monkey Ca, 254 ms; monkey He, 231 ms) (Fig. 1b). As attentional capture is in part defined by the rapidity of its associated behavioral response times, we sought to relate neuronal spiking activity to RT. We therefore segmented behavioral response times into quartiles at the session level (Fig. 1c), which we in turn related to the spiking activity measured in visual cortical area V4 (Fig. 1d). This task was designed such that a single stimulus of the array was present within the receptive field (RF) of the V4 multiunit whose responses we recorded. That way, we can compare the spiking

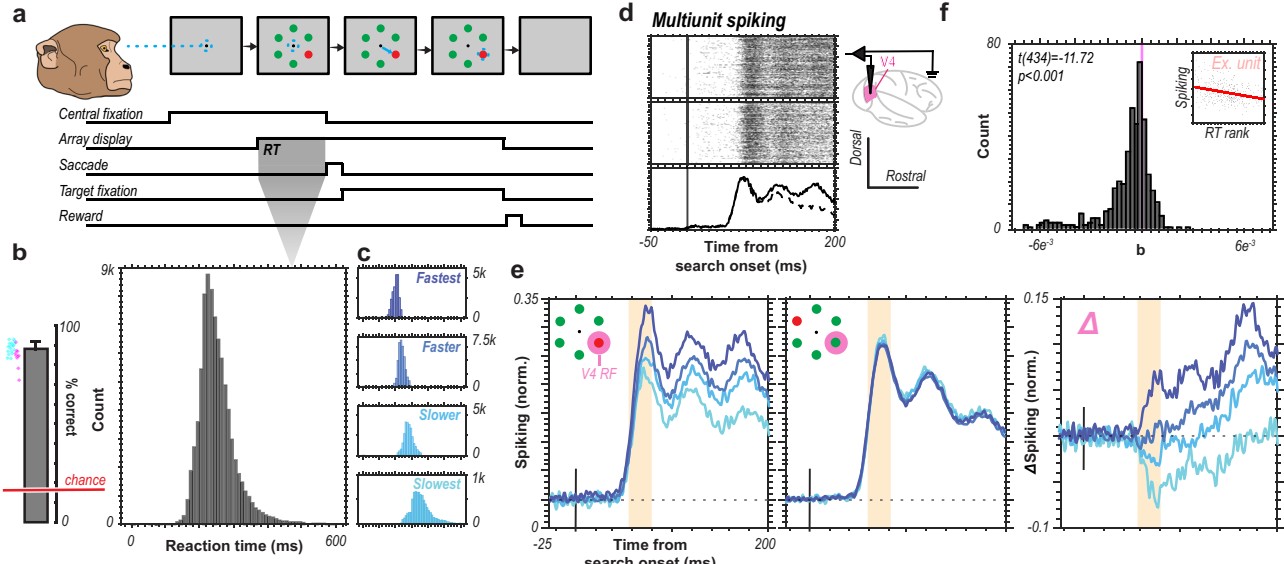

**Fig. 1 | Relationship between V4 spiking and response time during attention-capturing pop-out search. a** Monkeys performed a color-based pop-out visual search task to accurately identify the pop-out target as quickly as possible via an eye movement to the target. **b** Average accuracy within a session (n = 29 sessions, n = 2 monkeys) with an error bar denoting +2 SEM (left) and dots indicating individual session performance for both monkeys (monkey Ca, cyan; monkey He, magenta). Histogram of response times during correctly performed trials across all sessions (right). **c** Response times segmented in quartiles at the session level and color-coded for subsequent display. **d** Example V4 multiunit spiking (MUA) in response to a target presentation to the RF (top) or distractor presentation (middle) with their averages shown at the bottom (solid line, target; dashed line, distractor). **e** MUA spiking response to target (left) vs. distractor (center) presentation in RF averaged across all correctly performed trials and across units (n = 435) for each of the RT quartiles corresponding to **c**. Early component of the visual response highlighted in orange. Difference between target and distractor responses for each of the RT quartiles averaged across units (right). Early component of the visual response highlighted in orange. **f** Histogram of slopes of regressions for each unit of response time rank (within session) and unit spike rate across trials. Example inset shows scatter-plot and regression for one unit. Statistic reports the result of two-sided t test on the distribution of slopes.

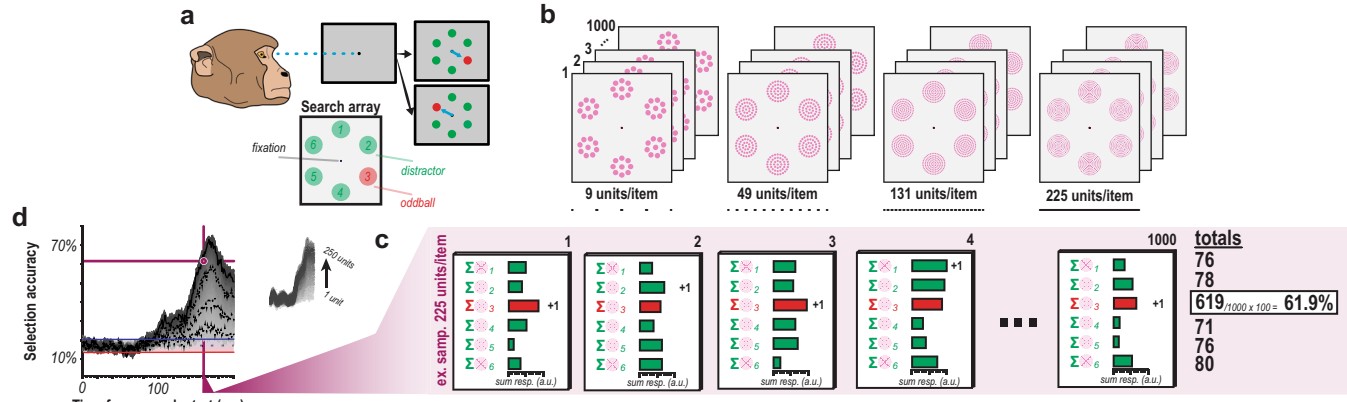

**Fig. 2 | Population reliability analysis. a** Monkeys performed 6-object color oddball search by making an eye movement to the oddball following presentation of the stimulus array. **b** Visualization of population reliability analysis, with magenta circles representing multiunits, not a different stimulus array, comprising the response to each stimulus. Four panels show example population sizes (9, 49, 131, 225) with stacking indicating the 1000 sampling simulations performed. **c** The summed multiunit response of each of the 6 (1 for each search item) samples was computed and their maximum was defined as selection of the associated stimulus. Each sample is a randomly selected set of trial-level multiunit responses for the corresponding stimulus type (i.e., oddball or distractor). This was repeated 1000 times for each population size for each millisecond across time to compute an oddball selection metric (percent of time oddball had the largest magnitude

response across sampling simulations). **d** Oddball selection as a function of time in pop-out search for population sizes 1–250 considering all correctly performed trials ($n = 1320282$ multiunit trial-wise responses) across both monkeys ($n = 2$ monkeys, $n = 29$ sessions). Values above the chance window indicate reliable population selection of the oddball. Chance is 16.67% for a 6-object search. Chance window is the empirically measured variability in selection accuracy in the absence of visual stimulation as computed as the 99% confidence interval during the baseline period (100 ms prestimulus epoch). Four example traces highlighted as black dashed lines correspond to the 4 example population sizes show in **b**. Crossing point of magenta vertical and horizontal lines, denoted by the magenta circle, indicate time and population size combination exemplified in **c**.

responses when the stimulus in the RF is the attention capturing oddball and when it is not. Qualitatively, it is apparent that where V4 multiunit responses to a distractor stimulus do not covary with RT (Fig. 1e, center) (Kruskal-Wallis Test, unit-wise average response 58–158 ms from search start: $H(3) = 1.04$, $p = 0.792$), there is the separation of the spiking responses when an oddball (attentional target) stimulus is present in the RF of the V4 multiunit (Fig. 1e, left) (Kruskal-Wallis Test, unit-wise average response 58–158 ms from search start: $H(3) = 7.91$, $p = 0.0479$). Moreover, this separation occurs early in the visual response (~60 ms following array onset). This response difference results in distinct target selection profiles (target response – distractor response) predicated on the RT quartiles (Fig. 1e, right). Speculatively, this effect, together with a marked tendency for multiunit responses to be lower on slower RT trials (Fig. 1f) suggest that behavioral outcome (RT) is somewhat predictable based on the earliest phase of V4 responses.

## Temporal evidence for feedforward selection

We quantified temporal evidence for feedforward selection via population reliability analysis. Population reliability analysis has previously been employed to derive selection times in a decision-making task[43]. Based on neurophysiological data, this analysis provides quantitative insights into both when a selection process is completed as well as the neural population size that is required for this selection to occur reliably. Briefly, population reliability analysis assumes there is a neural population representing each of the selectable objects or surfaces in a task. At each millisecond in time, we summed the activity of a randomly chosen population of multiunits to determine which item produced the largest population response. As we did not simultaneously record all 6 stimulated regions of V4, we instead employed sampling simulations. That is, we subsampled responses across trials representative of the varying stimulation conditions (i.e., oddball vs. distractor inside the receptive field). The stimulus yielding the largest response in this sample is taken as the selected item. This process is then repeated, each time taking randomly selected trial-level multiunit responses to the stimulus array to determine the frequency with which the oddball is selected. Crucially, this analysis can be performed over

time to determine when population-level selection for the oddball stimulus is significant relative to chance. This analysis also affords the ability to vary the number of multiunits included in the population to determine the requisite population size to detect population-level selection. We defined the oddball selection frequency computed by this analysis as our attentional capture metric. Values exceeding chance threshold indicate reliable, statistically significant attentional selection of the oddball stimulus (see Methods and Fig. legends for details on the statistical hypothesis tests).

To illustrate this analysis, consider an example calculation (Fig. 2). First, we determined the frequency with which the oddball is accurately selected 130 ms after presentation of the array for a population size of 225 multiunits (Fig. 2c, d). We assumed each of the 6 items is represented by the activity of 225 multiunits. We randomly selected 225 trial-level multiunit responses for each type of relevant stimulus. We summed those 225 multiunit responses for each item. The item with the largest summed response was tallied. We then repeated this process 1000 times using random (Monte Carlo) sampling. Of those 1000 samples, we counted the tallies for the oddball stimulus to find the frequency with which the oddball evoked the largest summed response (61.9% of the time, Fig. 2c). That provided 1 data point on the ordinate in the selection time plots (e.g., Fig. 2d). All other points were calculated by changing the time window (abscissa) and/or the number of multiunits (trace). In this example we do not separate trials by response time. As a result, we do not see selection for the oddball stimulus early in the V4 response. Figure 2d seems to indicate that oddball selection does not occur until about 100 ms after stimulus display, relatively long after the feedforward response of V4 neurons (50–60 ms). However, given the differentiation in V4 responses as a function of response time shown in Fig. 1e, f, we next quantified oddball selection as a function of reaction time.

We found that oddball detection varied in concert with reaction time (Fig. 3a–c). We see that for each of the quartiles, there is a distinct selection profile across the population sizes where accurate attentional selection exceeding the baseline variability (Fig. 3b, c, red and blue horizontal lines) occurs at different times in line with the differences in RT. That is, selection times (exceeding significance threshold)

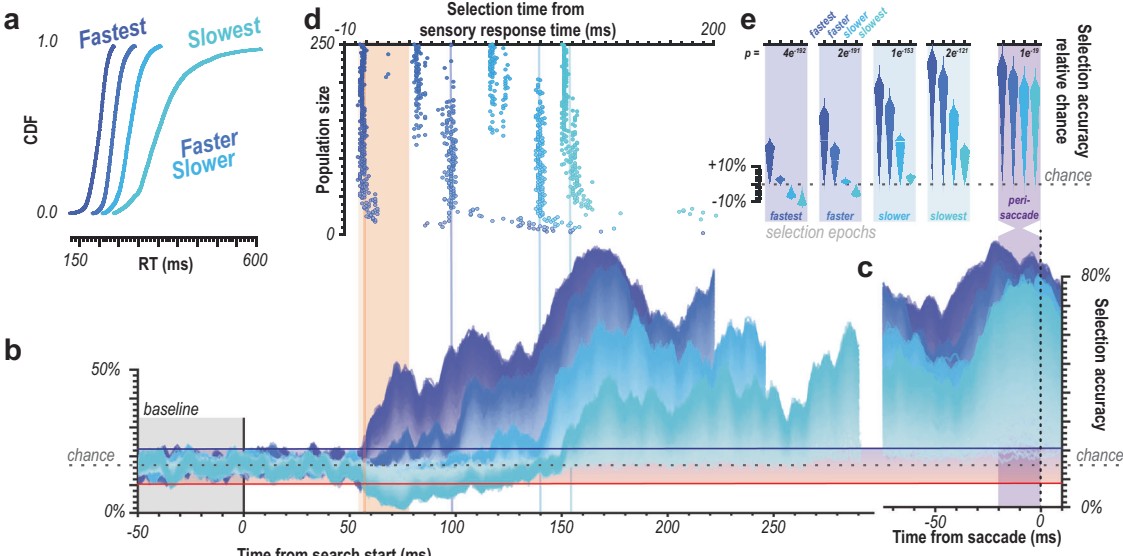

**Fig. 3 | Temporal evidence for feedforward attentional selection in sensory cortex. a** Cumulative density functions (CDF) of reaction times (RT) organized in quartiles ($n = 2$ monkeys, $n = 29$ sessions). **b** Oddball detection for each RT quartile. Data clipped 10 ms before each respective median RT. Orange highlights duration of the initial transient of sensory response. Population sizes 1–250 for each bin represented as lightest to darkest traces. **c** Oddball selection across time for each of the 4 quartiles, aligned on gaze shift to the oddball. Population sizes 1–250 for each bin represented as the lightest to darkest traces. Chance window is identical to that in **b**. **d** Time when each population size for each bin first exceeded chance threshold in **b**. Color indicates data from each RT quartile. Orange highlight indicates the duration of the initial transient of sensory response. Abscissa is relative to the average V4 visual response latency. **e** Oddball selection for each RT quartile immediately following each time where population 250 exceeded the chance window in each quartile as well as the window immediately preceding behavioral reaction time (far right). Background highlight indicates each selection epochs' corresponding quartile, which corresponds to the vertical line color passing through panels **b** and **d**. Violin plots are relative to chance (16.67%) and comprise the values from each population size (1–250). Statistic in the upper righthand corner of each subpanel indicates the result of one-way ANOVA between quartiles.

occur earlier on faster RT trials and later on slower RT trials, as indicated by the 4 vertical blue lines (Fig. 3b). We highlight the sensory response window in orange (Fig. 3b,d). This window is shown around the time the population spiking activity reaches 50% of its maximum firing (see Methods for details) and indicates when the first visual responses to the search array are occurring in V4. If there were a feedforward component to the selection, we should see differences in selection profiles as a function of RT during this time period. Significant detectable differences in population activation to oddball vs. distractor stimuli can already be observed in the earliest (50–60 ms following visual stimulation) sensory responses (Fig. 3d), even in small population sizes. Crucially, we observed significant differences in the attentional capture metric across four successive epochs (bins) of the feedforward response (Fig. 3e). This is seen by comparing the 4 violins within each epoch. This observation indicates that while the initial response to the oddball may not always evoke the largest population response, it entails sufficient information to predict the associated reaction time.

Next, we investigated whether the feedforward sensory population responses could predict response time (Fig. 4). To do so, we divided the response times further into 25 bins (24 used, slowest bin eliminated as outliers) (Fig. 5a). We again focus on the 20 ms following the sensory response latency (58–78 ms following array onset) as to determine the role of feedforward activation limit any potential contribution of feedback activity. No systematic differences in distractor responses were observed with reaction time; however, oddball response covaried nonlinearly with reaction time (Fig. 4b). We performed Bayesian modeling to determine whether the relationship was significant, with reaction time as the dependent variable and feedforward population spiking response as the independent variable (Fig. 4c). We used a 20 ms bin for consistency, further exploration of bin sizes and offsets are shown as well (Fig. 4d, e). This analysis revealed significant predictive value in the independent variable's (i.e.,

feedforward oddball response) coefficient estimates ($r$: M = −0.73, 89% CI = [−0.71, −0.74]; $\beta$: M = 4.90, 89% CI = [4.68, 5.15]), explaining a large fraction of the variance ($R^2 = 0.62$).

## Laminar evidence for feedforward selection

We next evaluated laminar evidence. The canonical columnar microcircuit details layer-specific activations for feedforward vs. feedback computations[5–7]. These patterns are robustly observed in sensory cortex[30,38,39,44–49] (Fig. 5a). Differences in granular layer (L4) synaptic activation as a function of oddball vs. distractor stimulation thus would indicate feedforward oddball signaling. Analyzing the fastest reaction time trials, we indeed observed a significant difference in synaptic activity L4 as a function of oddball vs. distractor presentations to the column's population receptive field (Fig. 5b) happening at the time of the stimulus-evoked response (Fig. 5c). This result indicates differences at the level of feedforward input into V4. We quantified this relationship using the modeling techniques used for spiking data, listed above (Fig. 5d). We again found a significant relationship between L4 feedforward synaptic activation to the attention-grabbing oddball and reaction time ($r$: M = −0.56, 89% CI = [−0.54, −0.57]; $\beta$: M = 1.70, 89% CI = [1.64, 1.77]; $R^2 = 0.26$). It is worth noting the poorer fit of this relationship with CSD compared to the spiking data shown earlier. This could be due to CSD being a nosier signal than spiking, or due to another factor that was not quantified.

In further evaluating the laminar profile of oddball detection, we observed greater than chance detection during the initial response in the granular input layers for the fastest response trials (Fig. 5e). It is worth noting that the same early selection can also be seen in the upper, and to a lesser extent, lower layers of the cortex. This is perhaps unsurprising as any feedforward propagation of the selection signal should flow from the middle input layers through the remainder of the laminar microcircuit. Therefore, we promote the early selection in the middle layers as the more relevant finding in evaluating the prediction

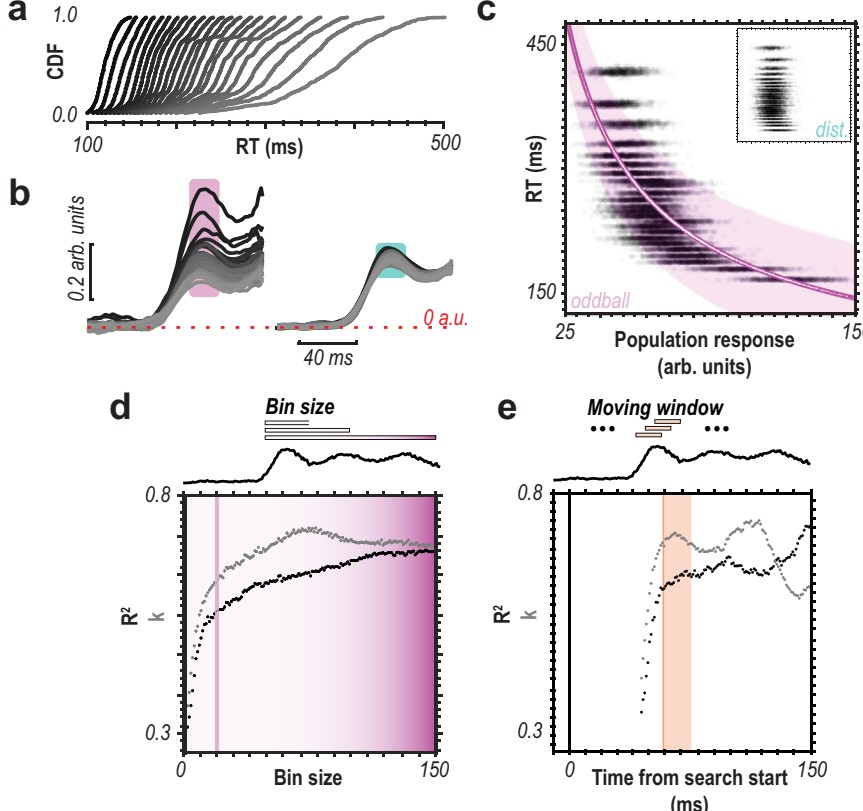

**Fig. 4 | Population spiking to target predicts response time. a** Cumulative density functions (CDFs) for reaction times (RTs) divided into 24 bins. **b** Multiunit responses to oddball (left) vs. distractor (right) corresponding to 24-bin RTs. Darkest traces are fastest trials, lightest are the slowest. Traces start at visual display. **c** Bayesian modeling 24-bin data with population size 250. RT as a function of population feedforward response (mean 58–78 ms postdisplay [20 ms bin]) to oddball, 1000 trials for each bin shown as black dots. Inset are distractor responses on identical scale. Magenta line shows the result of the power function fit. White data in the magenta line are median estimates for simulated trials. Magenta cloud is 89% credible interval of median estimates for fit. **d** Fit performance using different bin sizes for taking population response average. 20 ms was used in **c** and is indicated by a vertical line. Dot color refers to the data type, gray for the exponent value and black for the fit performance. **e** Fit performance moving 20 ms response average bin across time from search onset. 20 ms bin (starting at 58 ms) used in **c** highlighted with orange bar. Dot color refers to data type, gray for the exponent value and black for the fit performance.

of the feedforward hypothesis. Both the temporal evidence and the laminar findings here indicate a feedforward signature of attentional capture in the sensory cortex. In other words, the oddball stimulus gets emphasized over other stimuli during the initial volley of synaptic activation following stimulus onset. The underlying computation thus happens either at this moment and location, before that, or both. If the oddball detection occurred at a previous (upstream) location of visual processing, it thus must have been derived without feedback from area V4 or other downstream areas. Also note that the initial activation of V4 input layers precedes full sensory activation of earlier areas, such as V1 and V2[50], as well as the onset of distinguishable feedback responses in these areas[45]. This context further suggests that pop-out oddball detection occurs within the first wave of stimulus-evoked spikes[51] rather than during reverberant processing. Summarizing these findings, we find both temporal and spatial evidence (largely consistent across animals, Fig. 6) for the feedforward generation of a priority signal for attentional capture.

**Errant selection produces errant behavior**

An interesting secondary question is whether attentional capture in the feedforward response is entirely a factor of salience or if the information is relayed as a priority signal[14]. While priority signals have been described for frontal[52], parietal[53], and temporal[54] cortex, there has been no strong neurophysiological evidence for sensory cortical priority signals. We thus decided to test for the presence of priority signals in the sensory cortex. While incorrect behavioral responses are a small minority in pop-out search, the monkeys performed sufficient trials to yield a representative sample of error trials. However, this diminished sample restricts us from segmenting the response times into quartiles to evaluate potential differences that might contribute to differences in RT, as was done in the correct trials. Nonetheless, we can determine whether the population signal in V4 reflects the salience of the stimulus (which is constant between correct and error trials) or priority (which differs between correct and error trials).

Two alternative hypotheses emerge from this line of reasoning. If the feedforward response reflects salience, we expect robust attentional capture for the oddball, even when a distractor was (erroneously) selected as the target. If the feedforward response computes priority, however, we expect the population response to reflect the incorrect target selection[52,55]. We performed a population reliability analysis to distinguish between these two possibilities.

In line with the hypothesis that the feedforward sensory response represents a priority signal, V4 population responses selected (misidentified) the distractor, errantly capturing attention (Fig. 7a). Moreover, this selection was present in the initial response (Fig. 7b). Somewhat unexpectedly, we also found a small selection bias during the pre-display (fixation) period (Fig. 3c). Sufficiently large populations of units (e.g., >50 units) demonstrate ~2% bias in that baseline period. One could speculate that this observation suggests errant capture could be partially explained by modulated ongoing activity. However, this should be carefully considered, given the magnitude of this observation is quite small. In further pursuing this question, we

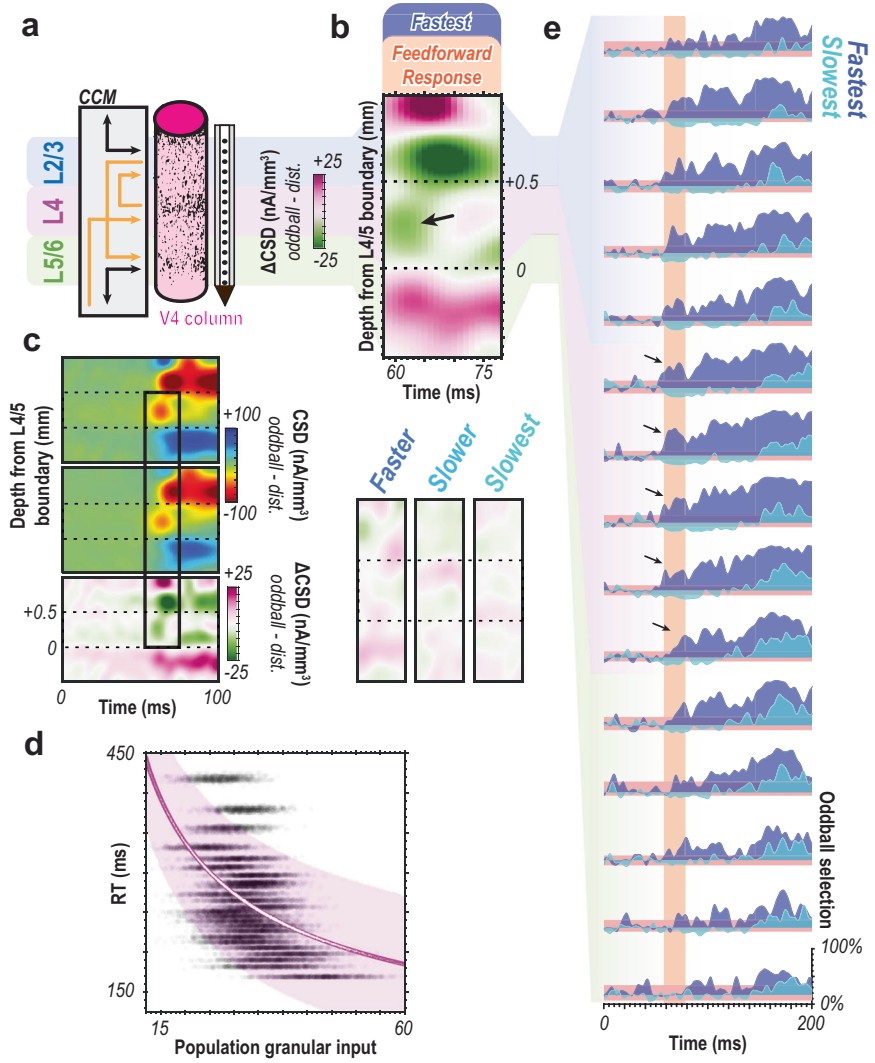

**Fig. 5 | Spatial evidence for feedforward attentional selection across the canonical cortical microcircuit. a** Cartoon illustrating the key laminar compartments (upper-L2/3, middle-L4, and deep-L5/6) with canonical cortical microcircuit (CCM) connectivity. Since earlier stages of sensory processing primarily project to L4 while feedback terminals innervate the layers above and below, feedforward computations should be indicated by differences in initial middle layer activation. **b** Laminar current source density (CSD, estimating synaptic activation) difference between oddball and distractor for the fastest, attention-capturing trials (top). Panel focuses on the 50 ms time window at the time of feedforward activation of the cortical column. Arrow indicates difference present in L4 where the oddball evoked a greater response. Plots underneath show CSD difference for the faster, slower, and slowest conditions from left to right with the same x, y, and z dimensions as above. No consistent CSD pattern is observed in these conditions. **c** Target (top) and distractor (center) stimulus-evoked CSD responses across all sessions for the fastest quartile plotted in line with associated difference plot (bottom).

Difference plot is same as **c** but on different timescale. Black outline highlights coincident oddball detection and stimulus-evoked responses. **d** Bayesian modeling of 24-bin data with population size 250 using putative synaptic currents in L4. Reaction time (RT) as a function of granular CSD magnitude (mean 58–78 ms post-display) to oddball, 1000 trials for each bin shown as black dots. Magenta line is the result of a power function fit. White data within magenta line are median estimates for simulated trials. Magenta cloud shows 89% credible interval of median estimates for fit. Slope indicates a negative relationship between L4 CSD magnitude to oddball stimulus and behavioral RT. **e** Oddball detection by cortical depth for fastest (dark blue) and slowest (cyan) bins relative to visual display for population size 250 using multiunit responses. Samples were localized to each depth (*n* = 15). Arrows denote significant oddball detection in the feedforward response in the middle layer. Orange highlights initial 20 ms window of feedforward visual response.

observed that errant capture was predominantly related to changes in synaptic activity in the deep layers of the cortex (Fig. 7d, e). Deep layers in V4 have been linked to behavioral output, and the difference found here is in line with that association[47].

**Prestimulus modulation of pertinent feature selective columns**
After noting the relationship between baseline activity and behavior in error trials (Fig. 7), we hypothesized that coordinated modulation of baseline activity (Fig. 8a, b) could bias capture. Previous work has implicated altered baseline activity in perceptual sensitivity to visual objects[11,16,17,56]. We thus structured the task to induce feature priming.

Specifically, we employed "priming of pop-out" to promote attentional capture of a specific feature, such as the red or green color (Fig. 8c). Searching for the same feature (e.g., red oddball among green distractors) repeatedly results in faster reaction times[57]. Swapping the target feature reinitiates priming for the new feature. This effect translates across species[58] and is also observed here (Fig. 8d). Neural correlates of attentional priming exist in frontal[40] and visual cortex[59]. Those previous reports show that V4 responses are related to RT, at least in evaluating differences as a function of priming[59]. However, these findings do not explain how feature representations are promoted for capture. This latter type of attentional priming can be

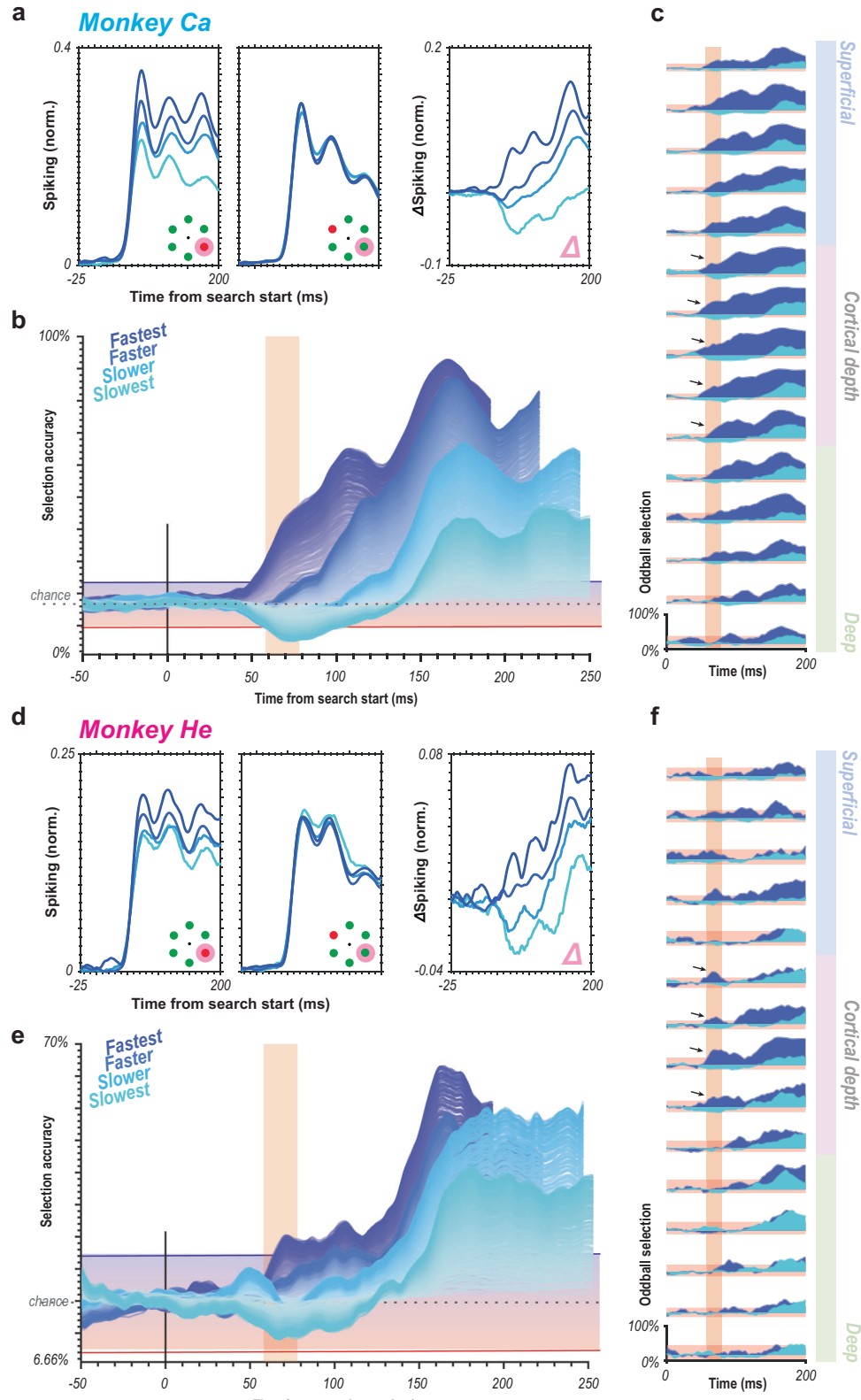

thought to reflect changes in the attentional prioritization of salient features.

We tested this by first identifying color-selective feature columns in V4. Topographic organization for color exists across V4[39,60,61] and can be observed at the cortical columnar level (Fig. 8a, b). We used a variation of the population reliability analysis to measure feature selectivity at the unit level. Briefly, a sample of 100 responses to red

and green stimuli in the unattended condition were taken for each unit 1000 times. We then took the sum of those samples and made a binary choice as to which color evoked a greater response. With 1000 iterations this yields the percent of time one color evokes a greater response than the other for a given unit. In Fig. 8a, b, percentages greater than 0 indicate selectivity for one color or the other. We found feature selective units and columns through these methods matching

**Fig. 6 | Evidence for feedforward attentional selection is consistent across animals. a,d** MUA spiking response to the target (left) vs. distractor (center) presentation in RF averaged across all correctly performed trials and across units for each monkey (a, monkey Ca, $n = 300$; d, monkey He, $n = 135$ units) for each of the RT quartiles. Difference between target and distractor responses for each of the RT quartiles averaged across units for each monkey (right). All traces smoothed for visualization (25 ms window, moving mean) **b,e** Oddball detection for each RT quartile for each monkey. Data clipped 10 ms prior to each respective median RT. Orange highlights duration of the initial transient of sensory response. Population

sizes 1–250 for each bin represented as lightest to darkest traces. Traces are smoothed for visualization only (25 ms window, moving mean), significance threshold (red-blue horizontal window) computed on unsmoothed data. **c,f** Oddball detection by cortical depth for fastest (dark blue) and slowest (cyan) bins relative to visual display for population size 250 using multiunit responses for each monkey. Samples were localized to each depth ($n = 15$). Arrows denote significant oddball detection in the feedforward response in the middle layer. Orange highlights initial 20 ms window of feedforward visual response. Data are smoothed for visualization only (25 ms window, moving mean).

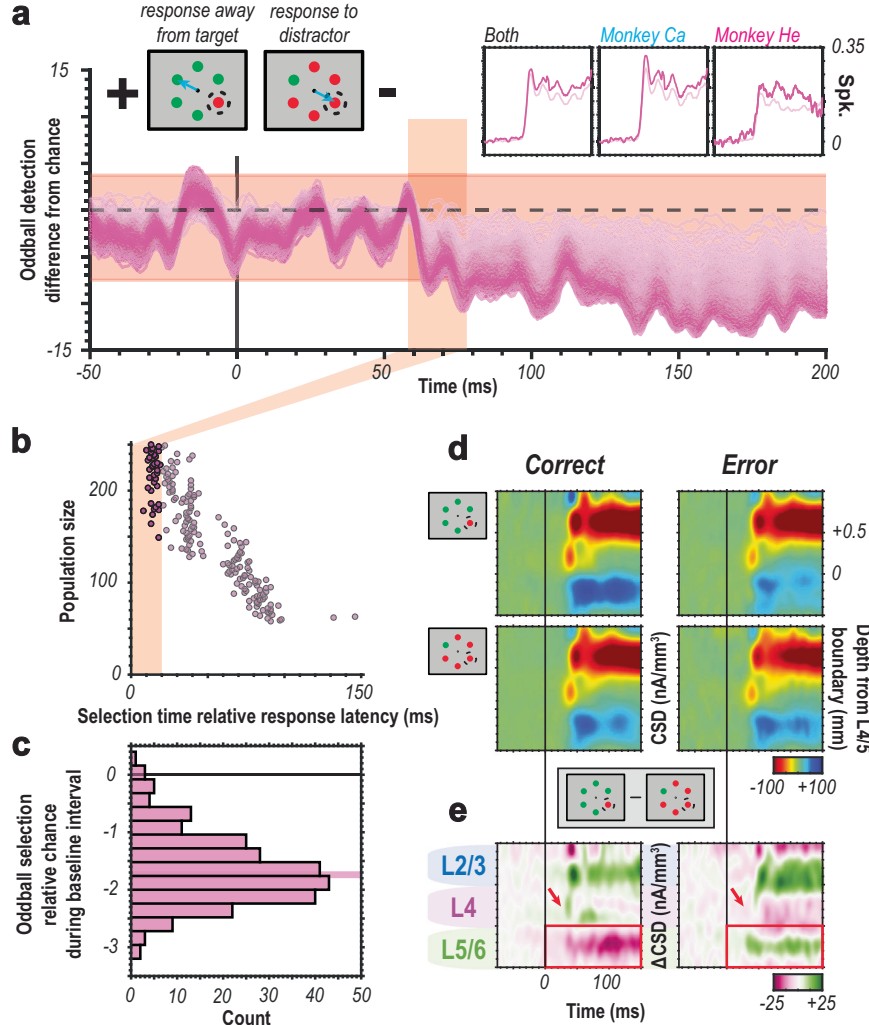

**Fig. 7 | Errant selection precedes errant behavior. a** Oddball selection for population size 1–250 (light to dark magenta) on incorrectly performed trials. Negative selection accuracy values indicate selection for the distractor. Feedforward visual response time window indicated in orange. Cartoon visualization of error types in the upper lefthand corner. Average normalized multiunit spiking for both error types (response to the distractor, darker magenta; response away from the target, lighter magenta) across both animals and in each shown in upper righthand corner and on same timescale as oddball selection plot. **b** Selection times for prioritized distractor for each population size. Feedforward response time window highlighted in orange. **c** Histogram of oddball selection during baseline period (50 ms prestimulus window). Small, but reliable bias in selection for misprioritized distractor observable before visual display (result of two sample t test

indicated in plot). **d** Comparison of putative synaptic activity for correct (left) vs. incorrect (right) trials across sessions for stimulus captured (top) vs. distractor (bottom) conditions. **e** Difference in putative synaptic activity between captured stimulus and distractor conditions for correct (left) vs. incorrect (right) trials. Green indicates either a stronger current sink during target presentation (e.g., superficial layer activity in the correct condition) or stronger current source in the distractor condition (e.g., deep layer activity in the error condition). Magenta indicates either weaker current source in target condition (e.g., deep layer activity in the correct condition) or weaker current sink in the target condition). Red arrows indicate notable difference in granular input sink and red box denotes observed difference in deep-layer putative synaptic source.

those reported previously with the same data, but different methods[39,49]. We measured the difference in baseline activity for the preferred vs. non-preferred feature before and after establishing behavioral relevance via priming (Fig. 8e). We found that baseline activity was significantly higher in columns selective for a color when

the color was behaviorally relevant (Fig. 8f). It is worth noting that the stimulus-evoked change from baseline for target vs. distractor is highly similar for both not primed and primed conditions. There does not appear to be a more complex interaction with the sensory response. The only difference of note is the persistent elevated activity. Note that

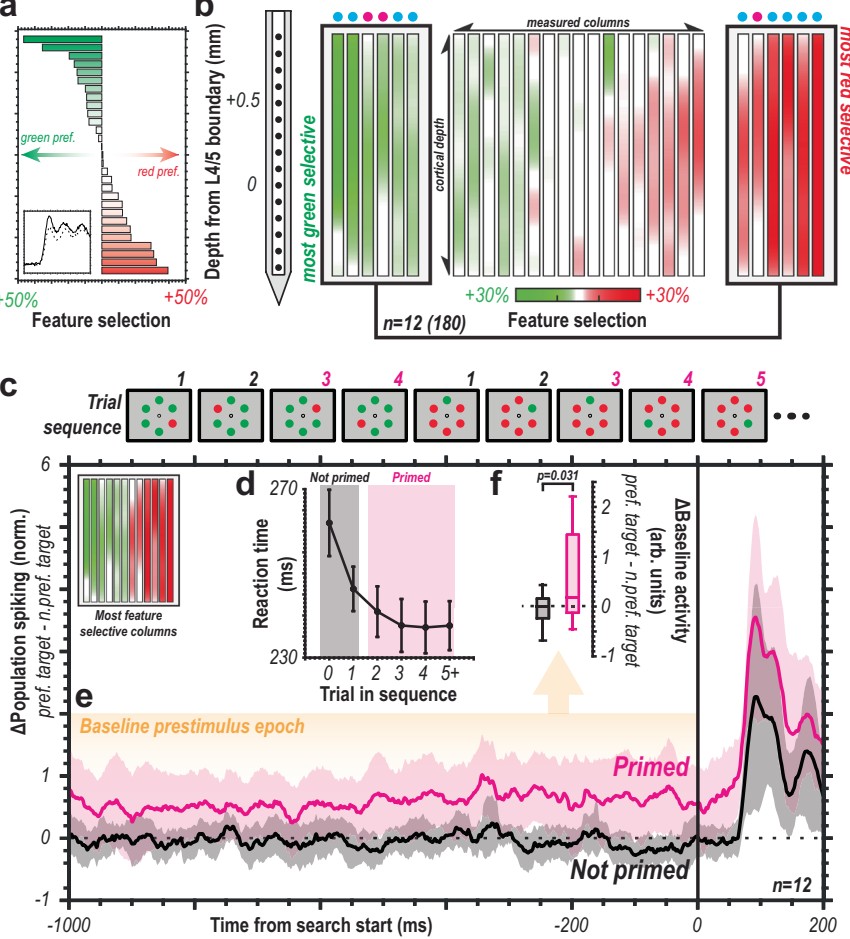

**Fig. 8 | Selection history modulates ongoing activity. a** Columnar feature selectivity organized from most green-preferring to most red-preferring columns ($n = 29$). Bar size indicates degree of feature selectivity across each column's multiunit spiking activity. Inset displays representative average red vs. green response for a red-preferring column (solid line, red response; dashed line, green response). Selectivity metric represents application of population reliability analysis at the individual multiunit level for selection of red vs. green. **b** Feature selectivity along depth for each column. Six most red- and green-preferring columns highlighted for further analysis. Colored dot above each column indicates monkey of origin (monkey Ca, cyan; monkey He, magenta) **c** Example trial sequence where target feature (red among green or vice versa) repeated before switching. This induced priming of target feature where primed (magenta) can be compared to not primed responses (gray). **d** Reaction time (medians with 95% confidence intervals) decreases with repeated search for a feature ($n = 29$ sessions). **e** Mean difference in baseline population spiking between preferred feature oddball and non-preferred feature oddball for not primed (black) and primed (magenta) trials for feature columns (inset top-left) ($n = 12$ columns, $n = 180$ multiunits). Clouds are 95% confidence intervals. **f** Boxplot of differences in baseline spiking for preferred feature target vs. non-preferred feature target for not primed vs. primed trials ($n = 12$ columns, $n = 180$ multiunits). Central mark indicates median, and the bottom and top edges indicate the 25th and 75th percentiles. Whiskers are +/−2.7 SD. Statistic is a two-sided t test.

this difference in firing persists in the absence of visual stimulation, thereby supporting the proposition of feedback modification in the case of task repetitions (priming).

### Feedforward selection does not require priming

With the understanding that feature-selective cortical columns can be modulated to promote their responses for attentional selection, we sought to determine whether feedforward selection was only present under priming conditions. To test this, we reduced the data to trials immediately following a switch in the priming sequences (e.g., the trial when the task switched from red among green to green among red). In these trials, the search target is not primed (and may in fact be negatively primed). We then repeated the Bayesian modeling of reaction time as predicted by oddball multiunit responses for these not primed trials. The procedure was identical to that described in Fig. 4. We found significant predictive value in the independent variable's coefficient estimates ($r$: M = −0.45, 89% CI = [−0.43, −0.48]; $\beta$: M = 1.73, 89% CI = [1.64, 1.86]) explaining a fraction of the variance ($R^2 = 0.18$). This finding indicates that even when feedback is promoting the feature representations associated with the distractors, the predictive relationship between oddball feedforward responses and reaction times is preserved. Thus, even though presentation of an oddball array leads to modulating feedback following task completion, this feedback cannot explain the feedforward selection we observed. In other words, feedforward attentional selection during pop-out is not predicated on prior feedback.

### Discussion

We found that the magnitude of neuronal population responses in sensory cortex to an attention-grabbing stimulus is predictive of reaction time and accuracy during attentional capture. Crucially, this relationship emerged in the earliest periods of the stimulus-driven response propagating feedforward. Moreover, oddball detection was observed in the initial current sink that marks the synaptic activity propagating through the granular input layer of sensory cortex. Remarkably, feature columns were tonically modulated by repeated task demands, adjusting ongoing activity to promote attentional capture for consistently pertinent features. In line with this notion,

errantly biased activation during the pre-stimulus epoch was associated with errant behavior. These findings demonstrate a role for sensory cortex in coordinating attentional priority. From a theoretical perspective, these observations resolve a long-standing debate on attentional capture[1–3]. Specifically, we find that a priority signal is automatically generated in a feedforward fashion. That signal is then used for tonic modulation via feedback to promote the detection of similar objects in subsequent searches. In other words, our attention is automatically captured by salient features. The speed of our behavior in response to these objects is dictated by the variability in their engendered sensory response. However, behavioral goals and historical context can influence which features we are more sensitive to, effectively promoting repeated attentional capture for objects comprised of those features.

It is interesting to relate these findings to those from the figure-ground literature. After all, the pop-out stimulus can be likened to an image that stands out from its background. Indeed, figure-ground segregation can happen rapidly and is seemingly in part feedforward and apparent already at the earliest stages of visual cortical processing in V1[62]. Laminar evidence is also consistent with this notion[63]. And while we can only speculate as to the exact mechanism producing the rapid attentional selection we observe in our data, the figure-ground literature supplies useful information that can inform future investigations. For one, it is plausible that the rapid selection we observe is mediated through horizontal connections within a given cortical area akin to figure-ground segregation[63]. Indeed, we see evidence for location-specific differences in V4 responses as a function of location relative to the pop-out stimulus. This finding might be related to the retinotopic, horizontal connectivity within V4 (Fig. 9). Perhaps, this rapid selection may even parallel the cell-type specific mechanisms that are observed in figure-ground segregation with distinct inhibitory cell types contributing more or less to the selection process[64]. It is also worth noting the seeming lack of flanking suppression, opposing findings in parietal[65] and frontal cortex[66,67]. However, this may be a factor of the distance between the target and its nearest distractor, something that should be investigated with more and less dense search arrays. Of course, all these mechanistic considerations for feedforward attentional selection are speculative. However, previous reports indicate a slight lag between the onset of the visual response and the salience response of figure-ground modulation in V4 (-15 ms), whereas our selection signal seemingly occurs simultaneously with response onset. It is important to note that the selection signal we document should not be considered a salience signal in the same way. Our findings indicate that RT can be predicted from these visual responses, but in many cases, the response to the oddball does not exceed that of distractors early on. In fact, it is sometimes lower, e.g., the slowest behavioral response time trials. It may therefore be more appropriate to omit consideration of this response modulation as a salience signal and treat it as simply the strength of the oddball response which has an impact the RT to that stimulus.

In evaluating population codes for the representation of sensory information, it becomes important to consider the size of a neural population that allows for certain computations to be performed[68]. For example, in behavioral tasks, an observation requiring an inordinately large population of neurons, could be inconsequential if that information cannot be relayed downstream for the execution of behavior. In our study, gaze must be redirected for our behavioral measure (reaction time) to occur. This process likely engages areas like the frontal eye fields (FEF)[69,70], which receive sensory information from V4[9,23,71–75]. Therefore, the population representation of the oddball stimulus must be relayed effectively from populations of V4 to FEF neurons. With this in mind, it is worth noting that reliable oddball detection at the population level can be observed in sets of 20 or fewer multiunits during the feedforward period (Fig. 3). Also, it seems that this feedforward selection is measurable when the population and

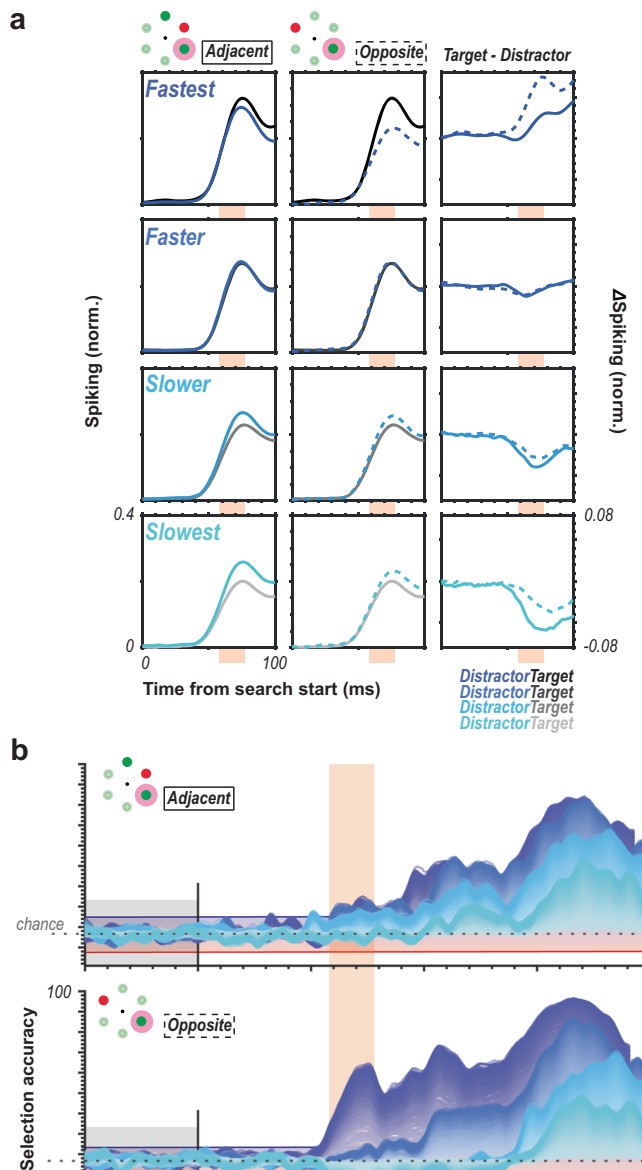

**Fig. 9 | Putative role for topographic organization in feedforward selection.** **a** Average target vs. distractor multiunit responses across all units in both animals ($n = 435$ units, $n = 2$ monkeys, $n = 29$ sessions) comparing when the distractor was positioned adjacent to the target (left, solid colored line), opposite the target (center, dashed colored line), as well as the difference between target and distractor responses for both conditions (right) for each of the 4 RT quartiles. **b** Oddball detection for each RT quartile for restricting the analysis to selection between target and adjacent distractors (top) and target and opposite distractor (bottom) conditions. Orange highlights duration of the initial transient of sensory response. Population sizes 1–250 for each bin represented as lightest to darkest traces.

temporal structure of a visual response is preserved (i.e., when we limit a population within the PRA to simultaneously recorded units on a given trial and preserve a degree of independence in the analysis) (Fig. 10). However, it would be interesting to revisit this question with simultaneous recordings of populations of neurons with different receptive fields. Together, this finding reinforces confidence that the detectability of this feedforward signal is not a confound of the pseudopopulation approach taken earlier, nor does it require an inordinately large population of neurons.

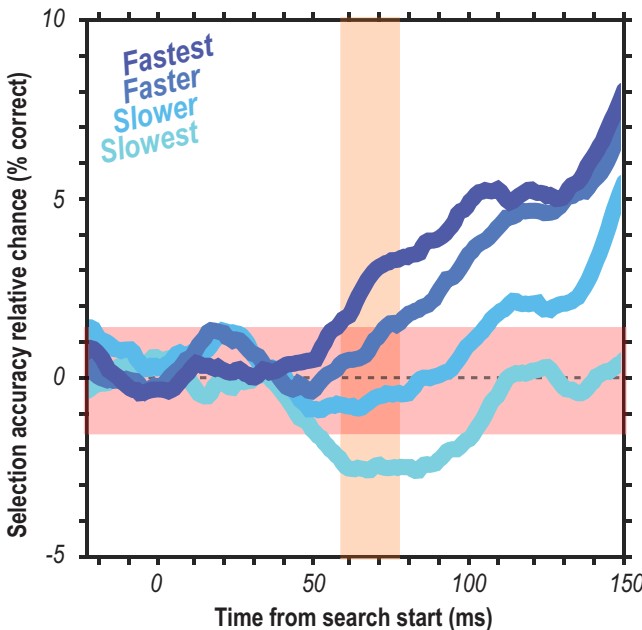

**Fig. 10 | Feedforward selection with preserved population activity.** Oddball target selection accuracy as a function of time for the 4 RT-quartiles across both monkeys ($n = 2$) and all sessions ($n = 29$). Population reliability selection accuracy was computed with only the combined multiunits of a single cortical column on a single trial thereby preserving the independence of population samples. Sampling was performed 1000 times for each quartile consistent with previous figures. Data are relative to chance (16.67% in 6-item visual search). Red highlights chance region computed from baseline variability. Orange highlights duration of the initial transient of sensory response.

It is also interesting to consider population size as a source for the variability of reaction time that is not explained by the modeling[76]. The different percentages of time the oddball is selected between traces suggest that the exact population size does impact the magnitude of detectability, at least in this population reliability metric. Therefore, we hypothesize that some variability observed in the behavioral response as a function of behavioral capture might be due to the size of the population propagating the signal. However, the questions remain, how is this controlled and is there a specific brain area responsible for this? One possibility is an accumulator of feedforward sensory information in a downstream brain area which appears biologically and computationally feasible[77].

In further considering the relationship between these feedforward responses and the ultimate behavioral response time: What aspect of neuronal function explains a range of several hundred millisecond for response times in such a simple task? One possibility is an aforementioned potential dependence on neural accumulators[78] which integrate information from upstream neurons to initiate the action (e.g., movement neurons in FEF[79]). Neurally-constrained models of such accumulators support such a mechanism[80–82]. Also notable, in examining the selection profiles of Fig. 3, it is interesting to note that the slowest response time trials yield initial negative deflection. This could indicate that there is an errant selective feedforward response that must be overcome in order to accurately identify the target. This finding, being a result of greater neural responses to a distractor than the target, is observable in other brain areas in parietal[65] and frontal cortex[55]. These slow trials could either require additional local processing or top-down feedback to override what would otherwise result in an incorrect response. This is seemingly corroborated in Fig. 7 where we see a negative deflection in the selection profile which persists instead of being "corrected". However, further investigation is necessary to elaborate what processing is undergone throughout this early

errant selection and how they are corrected in some instances (Fig. 3 slowest responses) but not always (Fig. 7).

And lastly, while we have found that modification of the priority signal exists tonically, is there an antecedent? That is, what instigates the persistent change in activity found in the behaviorally relevant feature columns? One hypothesis is that the initial presentation of the attention-grabbing oddball leaves sensory cortex in an altered state, more sensitive to the established pertinent feature[58]. Adaptation in sensory cortex can have potent effects[83–88], and is implicated in changing response characteristics at the level of cortical columns[89,90]. An alternative view might be that the frontal cortex regulates feature-based attentional modulation in the visual cortex; a candidate area (VPA) has been identified that could serve as a source[91,92]. Either hypothesis does not change the interpretation that the sensory cortex automatically computes attentional capture; however, resolving between them would provide insight into the minimum required neural circuitry to modify the priority signal.

## Methods

### Animal Care
Two male macaque monkeys (*Macaca radiata*; monkey Ca [age: 14 years], He [age: 12 years]) participated in this study. All procedures were in accordance with the National Institutes of Health Guidelines and the Association for Assessment and Accreditation of Laboratory Animal Care International's Guide for the Care and Use of Laboratory Animals, and approved by the Vanderbilt Institutional Animal Care and Use Committee in accordance with United States Department of Agriculture and United States Public Health Service policies. Animals were pair-housed. Animals were on a 12-hour light-dark cycle and all experimental procedures were conducted in the daytime. Each monkey received nutrient-rich, primate-specific food pellets twice a day. Fresh produce and other forms of environmental enrichment were given at least five times a week.

### Surgery
All surgical procedures were performed under aseptic conditions. Anesthesia was conducted with animals under $N_2O/O_2$, isoflurane (1–5%) anesthesia mixture. Vital signs were monitored continuously. Expired $PCO_2$ was maintained at 4%. Postoperative antibiotics and analgesics were administered while animals remained under close observation by veterinarians and staff. Monkeys were implanted with a custom-design head post and MR-compatible recording chamber using ceramic screws and biocompatible acrylic. A craniotomy over V4 was opened concurrent with the recording chamber.

### Magnetic resonance imaging
MR images for chamber localization and guiding of linear electrode penetrations perpendicular to the cortical surface were taken from anesthetized animals placed inside a 3 T MRI scanner (Philips). T1-weighted 3-dimensional MPRAGE scans were acquired with a 32-channel head coil equipped for SENSE imaging. Images were acquired using a 0.5 mm isotropic voxel resolution with the following parameters: repetition time 5 s, echo time 2.5 ms, and flip angle 7°.

### Identification of V4
Recordings took place on the convexity of the prelunate gyrus in approximately the dorsolateral, rostral aspect of the V4 complex, where receptive fields are located at about 2–10 degrees of visual angle (dva) eccentricity in the lower contralateral visual hemifield[93]. Laminar recordings took place at locations where the array could be positioned orthogonal to the cortical surface, as verified by MRI and neurophysiological criteria (i.e., overlapping receptive fields). Recording sites were also confirmed via histological staining by dipping the electrode arrays in diiodine prior to the final recordings in monkey He[59].

## Task design: Pop-out search

Monkeys viewed arrays of stimuli presented on a CRT monitor with 60 Hz refresh rate, at 57 cm distance. Stimulus presentations and task timing was controlled using TEMPO (Reflective Computing). Visual presentations were monitored with a photodiode positioned on the CRT monitor so that electrophysiological signals could be reliably aligned offline. Red (CIE coordinates: x = 0.648, y = 0.331) and green circles (CIE coordinates: x = 0.321, y = 0.598) were used as stimuli, rendered isoluminant to a human observer at 2.8 cd/m$^2$ on a uniform gray background. As we are limited to two colors and cannot account for potential differences in perceived brightness between macaques, we qualify our two stimuli as distinct 'features' at the intersection of color and luminance information. Nonetheless, we report the colors used in this study for the ideal human observer. Cone excitation was computed from the CIE coordinates and luminance[94]. The following cone excitations were measured for red: $\varepsilon_L = 2.37$, $\varepsilon_M = 0.43$, $\varepsilon_S = 0.0014$; green: $\varepsilon_L = 1.74$, $\varepsilon_M = 1.06$, $\varepsilon_S = 0.0030$; and the background: $\varepsilon_L = 1.86$, $\varepsilon_M = 0.94$, $\varepsilon_S = 0.023$. Cone contrasts for red stimulus were found to be: $C_L = 0.27$, $C_M = -0.54$, $C_S = -0.94$; and the green stimulus: $C_L = -0.06$, $C_M = 0.13$, $C_S = -0.87$.

Trials were initiated when monkeys fixated within 0.5 dva of a small, white fixation dot (diameter = 0.3 dva). The time between fixation acquisition and array presentation varied between 750–1250 ms, taken from a nonaging foreperiod function to eliminate any potential effect of stimulus expectation[95–97]. Following the fixation period, the stimulus array consisting of 6 items was presented. Stimuli were scaled with eccentricity at 0.3 dva per 1 dva eccentricity so that they were smaller than the estimated V4 receptive field size (0.84 dva per 1 dva eccentricity[98]). The polar angle positioning of the items relative to fixation varied from session to session so that one item of the stimulus array was positioned at the center of the population receptive field under study. Items were spaced such that only one item was in the V4 receptive field, with uniform spacing in polar angle and equal eccentricity.

Monkeys engaged in a search task while viewing the stimulus array. One item in the array was a different feature (red or green, respectively) from the others. Position of the oddball on each trial was randomly chosen with equal probability for any of the positions (16.6%). Monkeys earned fluid reward for shifting gaze directly to the oddball item within 1000 ms of array presentation and maintaining fixation within a 2–5 dva window around the oddball for 500 ms.

Eye movements were monitored continuously at 1 kHz using an infrared corneal reflection system (SR Research). If the monkey failed to look at the oddball, no reward was given, and a 1–5 s timeout ensued. Trials were organized into blocks such that the animal searched for the same target feature for 5–15 repetitions. Target feature remained the same, but target location varied randomly. Completing the block resulted in the target and distractor features swapping.

## Neurophysiology

Laminar extracellular voltages were acquired at 24.4 kHz resolution using a 128-channel PZ5 Neurodigitizer and RZ2 Bioamp processor (Tucker-Davis). Raw signals were output between 0.1 Hz and 12 kHz. Data were collected from 2 monkeys (left hemisphere, monkey Ca; right hemisphere, He) across 70 recording sessions (n = 31, monkey Ca; n = 39, He) using 32 channel linear microelectrode arrays with 0.1 mm interelectrode spacing (Plexon). Each recording session, electrode arrays were introduced into the prelunate gyrus through the intact *dura mater* using a custom micromanipulator (Narishige). Electrode arrays were positioned so they spanned all layers of V4 and had a subset of electrodes positioned outside of cortex. 29 (n = 20, monkey Ca; n = 9, He) of 70 sessions were included in the final analysis. The remaining 41 sessions were found to either not have a discernable CSD profile for laminar alignment, not be orthogonal to the cortical surface, or not have enough priming blocks for the feedback mechanism

analysis and were thus removed from analysis. Analysis of neurophysiological data was done using MATLAB R2020b (The Mathworks) and R v4.1.2 (R Foundation for Statistical Computing) with RStudio v2021.09.2 and RStan 2.21.1.

## Receptive field mapping

To determine the orientation and eccentricity of the visual receptive fields, monkeys performed a receptive field mapping task prior to the main task. Monkeys fixated for 400–7000 ms while a series of 1–7 stimuli were presented that spanned the visual field contralateral to the recording chamber. Stimuli were 5 high-contrast concentric white and black circles that scaled in size with eccentricity (0.3 dva per 1 dva eccentricity). In all recording sessions, stimuli could appear in a random location. These random locations spanned the lower visual quadrant contralateral to the recording chamber. Location spacing was in 5° angular increments relative to fixation and in eccentricities ranging from 2 dva to 10 dva in 1 dva increments. Each stimulus was presented for 200–500 ms with an interstimulus interval of 200–500 ms. If the animal maintained fixation for the duration of the stimulus presentation sequence, they received a juice reward. During this receptive field mapping task, multiunit activity, gamma power (30–90 Hz), and evoked local field potentials (LFPs, 1–100 Hz) were measured across all recording sites. Online, we measured the response across visual space for each recording site. Recordings proceeded to the feature search task if there was qualitative homogeneity of receptive fields along depth. Receptive field overlap for these data have been reported previously[39]. The receptive field center was chosen to be the stimulus location that evoked the largest response along the depth of recording sites. Following receptive field identification, the stimulus array in the feature search task was then oriented so that its eccentricity coincided with the location of the receptive field (eccentricity: 3–10 dva) and a single array item was placed at the center of the receptive field (size: 0.9–3 dva).

## Identification of cortical laminae

Positions of the individual recording sites relative to the layers of V4 were determined using current source density (CSD) analysis. CSD reflects an estimate of local synaptic currents (net depolarization) resulting from excitatory and inhibitory postsynaptic potentials[46]. CSD was computed from the raw neurophysiological signal by taking the second spatial derivative along the electrode contacts[48,90,99–101]. CSD activation following presentation of a visual stimulus reliably produces a specific pattern of activation which can be observed in primate visual cortex[48,100], including V4[30,39,49,102,103]. Specifically, current sinks following visual stimulation first appear in the granular input layers of cortex, and then ascend and descend to the extragranular compartments. To compute the CSD from the LFP, we used previously described procedure:[99]

$$CSD(t,d) = -\sigma \left( \frac{x(t, d-z) + x(t, d+z) - 2x(t,d)}{z^2} \right) \quad (1)$$

where the CSD at timepoint t and at cortical depth $d$ is the sum of voltages $x$ at electrodes immediately above and below ($z$ is the interelectrode distance) minus 2 times the voltage at $d$ divided by the interelectrode-distance-squared. That computation yields the voltage local to $d$. To transform the voltage to current, we multiplied that by -$\sigma$, where $\sigma$ is a previously reported estimate of the conductivity of cortex[104]. For each recording session, we computed the CSD and identified the initial granular layer (L4) input sink following visual stimulation. Sessions were aligned using the bottom of the initial feedforward input sink as a functional marker. We defined the size of individual laminar compartments uniformly relative to space. Throughout, 'middle' refers to the estimate of the granular input layer 4 (0.5 mm space above the CSD initial sink functional marker), 'upper' refers to

the estimate of supragranular layers 2 and 3 (0.5 mm space above the L4 compartment), and 'lower' refers to the estimate of infragranular layers 5 and 6 (0.5 mm space below the L4 compartment).

## Population spiking

Spiking activity at the level of multiunits was used for control analyses as it reliably reflects neural population dynamics[105]. Detection of multiunit activity was achieved through previously described means[106]. This method has proved useful across brain areas and research groups[59,63,107–111]. Briefly, broadband neural activation was filtered between 0.5–5 kHz, the predominate range of spiking activity. The signal was then full-wave rectified and filtered again at half the original high-pass filter (0.25 kHz) thereby estimating the power of the multi-unit activity. For filtering, we used a 4th-order Butterworth filter. Spiking responses were baseline corrected by subtracting the average activity in the 100 ms window preceding visual display onset at the trial level. This baseline correction was not performed for the feedback analysis. Spiking was normalized at the trial level with a z score method where the standard deviation was taken as the standard deviation of the baseline period activation in the 100 ms window before stimulus presentation.

## Feedforward sensory response window

Determining the implications of the feedforward response to attentional capture required accurate identification of the timing of said feedforward response. Here, the window of the feedforward response is defined as the 20 ms following the time at which the mean population spiking response first reaches 50% of its maximum response. This definition yielded a response latency of 58 ms (mean=58, 95% confidence interval = [57,59]) with the window being defined as 58–78 ms following visual display onset.

## Sorting responses by reaction time

Several analyses were conditioned on sorting trials by behavioral reaction time. Through this procedure, trials were rank ordered by reaction time from fastest to slowest on a session-by-session basis. For example, if session $n$ contained 2000 trials, each trial was ranked from 1 to 2000 by reaction time, and then normalized as a percentile. This ranking was completed individually for each session so that individual sessions could be sampled equally for each binned condition. Two binning procedures were performed, one coarse (4 bins) and one fine (25 bins). For the fine-binning-conditioned analyses, the slowest bin (slowest 2% of trials) was omitted from analysis as outliers in otherwise efficient, pop-out search.

## Population reliability analysis

Population reliability analysis was used to establish whether and when populations of V4 neurons selected an attention-grabbing oddball[43]. Crucially, this analysis estimates when selection occurs in time as well as how many neural units are required for this selection to occur reliably. This analysis is performed by simulating trials using data from the entire population of multiunit responses across all sessions. Each simulated trial is defined as an event where a behavioral response must be made to a stimulus with multiple alternatives present. For this computation, each alternative is represented by the response of a distinct neural population with a predetermined size. We varied the population size between 1 and 250. In this study, the search task contains six alternatives, thus we estimated a population response for each of the six stimuli. The population response is defined as the sum of responses of each of the sampled responses – where each response is an empirically measured trial-level multiunit response to the stimulus germane to the alternative's response that is being estimated. Therefore, we chose five distinct, randomly sampled population responses to distractor 'alternatives' and one to an oddball 'alternative'. For each point in time within each simulated trial, we measured

which alternative provoked the highest response magnitude. This selection metric represents our estimated priority signal. By simulating more and more trials (n = 1000 for each computation in this study), we computed the percent of time each alternative is selected at each timepoint during a simulated trial. Here, we were specifically interested in the percent of time the oddball was selected by this measure. We defined this percentage as the selection accuracy in identifying the behaviorally relevant oddball. For six objects, chance (selection invariability between population responses) was calculated to be 16.67%. We computed empirical selection bounds to estimate when the selection accuracy exceeded chance by measuring the variability in selection accuracy for all 1–250-unit populations during the baseline, prestimulus display, epoch and setting the thresholds to the 99% confidence interval.

## Bayesian modeling

Bayesian modeling was performed using Stan through the RStan interface. Sampling was done using Markov chain Monte Carlo (MCMC) methods with the following parameters: chains, 4; warmup samples, 2000; total samples, 5000; thinning, 2. A power function fit was assessed. The outcome $RT_i$ (reaction time on simulated trial $i$) can be modeled as:

$$\mathrm{RT}_i | \alpha, \beta, r, \mathrm{SPK}_i \sim N(\phi_i, \sigma_e) \tag{2}$$

where:

$$\phi_i = \beta * \mathrm{SPK}_i^r + \alpha \tag{3}$$

Reaction time (RT) for simulated trial $i$ is modeled as the population multiunit spiking activity (SPK) for simulated trial $i$ with coefficient $\beta$. $r$ is the exponent for the power function fit. Population spiking activity was defined as the sum of activity across the population. In supplementary analyses we also explored the same relationship to reaction time, but with the magnitude of the granular input sink taken as the average of 5 recording channels immediately above the L4/5 boundary for each trial. We set minimally informative priors for the power function fit as: $\alpha \sim \mathrm{LogNormal}(0, 0.5)$, $\beta \sim \mathrm{LogNormal}(1, 0.5)$, $r \sim \mathrm{Gamma}(1, 3)$, $\sigma_e \sim \mathrm{Gamma}(0.5, 5)$.

From this modeling, we computed median estimates for each simulated trial as well as the associated 89% credible intervals. We also computed reaction time estimates for the range of population spiking responses observed using the median estimate of each coefficient. In evaluating the posterior distributions, we were interested in the median (M) estimates for the coefficients $\beta$ and $r$ as they reflect the predictive utility of the independent variable (population spiking response). In particular, non-zero $\beta$ and $r$ different than 1 indicate significant utility provided the 89% credible intervals (89% CI) for those estimates do not include their respective non-predictive values.

## Feature selectivity

Feature (red vs. green) selectivity was derived from population spiking observed along recording sites. Responses were taken when a red stimulus was presented to the receptive field of the cortical column and when a green stimulus was present in the receptive field. We employed a two-alternative version of the population reliability analysis to estimate the selectivity of each multiunit for red vs. green. For each multiunit we took 100 red and 100 green stimulus presentations (effectively population size 100) 1000 times (bootstrapped simulated trials) for the reliability analysis. Specifically, we took the average response 60–160 ms following visual display. This yielded a selection accuracy metric for red vs. green where deviation from 50% chance indicated preference for one color or the other. Presence of feature selective columns in this dataset was confirmed in previous reports[39,49].

**Reporting summary**

Further information on research design is available in the Nature Portfolio Reporting Summary linked to this article.

## Data availability

The data generated in the study have been deposited in the Data Dryad database accessible through the https://doi.org/10.5061/dryad. s1rn8pk9t.

## Code availability

Specialized code for the sampling simulations and Bayesian modeling is freely available at: https://github.com/westerberg-science/ attentional-capture-code. The original release of this code can be found through the https://doi.org/10.5281/zenodo.8163978.

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

## Acknowledgements

The authors would like to thank and G. Bahg, G. Cox, S. Lilburn, G. Logan, T. Palmeri, and J. Theeuwes for their comments. This work was supported by the National Eye Institute [grant numbers: R01EY027402 (A.M.), R01EY019882 (J.D.S., G.F.W.), R01EY008890 (J.D.S.), P30EY008126 (J.D.S., G.F.W., A.M.)] and the Natural Sciences and Engineering Research Council of Canada [grant number: RGPIN-2022-04592 (J.D.S.)]. J.A.W. was supported by fellowships from the National Eye Institute [grant numbers: F31EY031293, T32EY007135] and the International Human Frontier Science Program Organization [grant number: LT0001/2023-L]. Imaging support was provided by the Vanderbilt University Institute for Imaging Science through a grant from the National Institutes of Health Office of the Director [grant number: S10OD021771]. Supercomputing resources were provided by the Vanderbilt University Advanced Computing Center for Research and Education.

## Author contributions

Conceptualization, J.A.W., J.D.S., G.F.W., A.M.; Data Collection, J.A.W.; Formal Analysis, J.A.W.; Data Visualization, J.A.W.; Original Draft, J.A.W.; Revisions and Final Draft, J.A.W., J.D.S., G.F.W., A.M.

## Competing interests

The authors declare no competing interests.
