## [Peer Review File · Nature Communications]

Feedforward attentional selection in sensory cortexREVIEWER COMMENTS

Reviewer #1 (Remarks to the Author):

This paper describes a study into the neuronal nature of attentional capture. Using laminar probes, the authors can distinguish responses from different layers of the cortex and make inferences about the likely feedforward and feedback contributions to the observed responses. They see signatures of attentional capture in the earliest part of the response and in layer 4 and conclude that attentional capture is generated in a feedforward fashion.

The question this manuscript addresses is interesting. The nature of attentional capture has been debated for decades and insight into the neuronal basis provides valuable information to this discussion. This study appears to provide such information. However, the manuscript is very dense which makes it difficult to properly evaluate the strength of the evidence that is provided for the conclusions. Selection choices in several analysis steps are not clearly explained in the main text. I think the argument could be strengthened a lot by a more explicit and extensive description of the analyses and specific conditions that are used to support the conclusions.

Some mechanistic interpretations of the results are stated as conclusions whereas they seem more like discussion points to me (see further comments for details).

Since the current work contrasts two hypotheses about the potential underlying mechanisms of attentional capture (feedforward vs feedback), I would expect these theories and their predictions for neural mechanisms to be described with a bit more detail in the introduction.

In the introduction, the authors state their results imply that the 'neural computations underlying attentional capture' occur within the earliest responses of V4 neurons. Here, I think they are speculating, as their data show a neural signature or correlate but not an actual computation. I'd suggest rephrasing these claims to state that these computations will have had to occur before or at the level of the early V4 response.

While I appreciate the in-line figures in this manuscript, I do think the figures are a bit too dense and small, especially figure 1. Is this due to a limit on the number of figures? If not, I think it would be nicer if Fig 1 would be split into two figures: one with the info from Fig1A-E, and one with the RT-dependent analyses explained in Fig1F-M. In addition, perhaps put panel B (task & stimuli) before A (responses split by task/stimulus) and I don't think panel C helps explaining the population reliability analysis (PRA)

much. I first thought there was something different about the stimuli before realizing it's merely a visualization of the number of responses/units that are included in the analysis.

Another problem with the panels in Fig 1 is the different selection criteria for the data in the panels based on RT, population size, bin of the PRA, etc. Altogether, it takes a very long time for the reader to parse figure 1 and its meaning for the result claims. This should be explained more extensively.

The study reports on data from two monkeys but all results are pooled. Are the conclusions supported by results in both animals? In other words, are the same result patterns observed in individual monkeys?

In Figure 1f,i the difference between the colors of the two fastest response groups is difficult to see. In Fig 1i the time is relative to the sensory response time. How is this determined?

The MUA traces for responses at different RT classes are not shown, only the 'selection accuracy' traces of the population reliability analysis. Since this is a derived signal, it would be good to also show the average oddball vs distractor traces for these same classes (as in Fig 1a). This holds for other analyses in the paper as well. I appreciate the PRA but feel it is more of secondary analysis after showing the MUA timecourses and not the only thing to show.

To what extent does the relationship between the RT and population response (Fig 1m) depend on the chosen bin. Here 58-78 ms is chosen. The method section describes how this window was determined but in the main text this should also be briefly explained.

The laminar analysis feels under-reported. It is only done on the fastest RT trials. Are similar patterns present in slower trials as well? Furthermore, the authors focus on the initial volley of activity in layer 4, but similar 'bumps' can be seen in layer 2/3 and (to a slightly lesser extent) layer 5/6 as well. These are not really discussed. What do they mean for the feedforward hypothesis?

In the analysis of the error trials, responses are now not split by RT and only performed on an early time window of the response, correct? I suppose it is more difficult to do more fine-grained analysis here because of the fewer error trials, but these differences in inclusion criteria should be more clearly described and motivated.

In the priming analysis, can the authors comment on the stimulus response above baseline for the primed vs unprimed oddballs? With the raised prestimulus baseline, the response increase to the stimulus seems comparable for primed and unprimed cases.

For the section 'Ffw selection does not require priming', what time window is used to determine spiking responses for the Bayesian modeling and why (same goes for Fig 1)?

In the discussion, the authors state that their findings indicate that sensory cortex 'dictate attentional capture'. This might be true, but cannot be conclusively concluded from the fact that signatures of attentional capture are present in the early response of neurons in feedforward layers. In addition, there is no evidence that 'this signal is used for tonic modulation via feedback'. Any other signal in the chain from sensory code to response could also drive this feedback. The mechanism can definitely be discussed but right now it is stated as fact, which seems a step too far.

The links to code and data currently do not yet go anywhere. I suppose this will be made public upon acceptance/publication, but would like the authors to comment on that.

Reviewer #2 (Remarks to the Author):

Westerberg et al. investigate the influence of pop-out and priming-of-popout on the activity of neurons in area V4 and how this activity relates to the reaction times of monkeys trained to make an eye movement to the pop-out stimulus. This study is a valuable addition to the literature on visual search and pop-out. However, I do have a number of concerns that need to be addressed before I can recommend acceptance of the Ms.

Major points:

1) I would like to see a direct comparison between the feedforward response, e.g. visible in Fig. 1a and the response modulation. My impression is that the modulation comes later than the feedforward response, i.e. that seems to be visible in Fig. 1a. If this is indeed the case, i.e. the modulation occurs later than the feedforward response, then the conclusion that "[...] this relationship emerged in the earliest parts of the stimulus-driven response" (first sentences of the discussion) is not supported by the data. To address this point, the authors should directly compare the time-course of the visually driven response to the time-course of the pop-out modulation. I suspect that the visual response will precede the pop-out modulation.

- Same for the CSD: can the authors directly compare the time-course of the stimulus driven CSD to that caused by pop-out?

- The authors defined the window of the feedforward response as the 20 ms following the time at which the mean population spiking response first reaches 50% of its maximum response. That seems to imply that the earliest spikes, the ones that are really feedforward are not considered.

- The authors may want to relate their results to those of Poort et al. (Neuron, 2012) demonstrating that the V4 visual response precedes a salience response driven by figure-ground segregation (which is related to the pop-out signal described here) by about 15ms.

- If the effect is indeed visible in the feedforward response, i.e. in the very first spikes, the reader wonders where the information, that requires a comparison between the stimulus in the RF and the other stimuli can come from. In other words, what is the signal path from the stimuli outside the RF to the V4 neurons that is short enough to influence the first V4 spikes, which are presumably driven from the V4-RF itself?

2) It remains unclear whether the effects were replicated in both monkeys. The resampling analysis seems inappropriate from a statistic point of view. The reason is that the resampling approach considered trials to be statistically independent, irrespective of whether they come from one or the other monkey (and recording session – although that presumably plays a smaller role here). Sampling from all the trials and recording sites will obscure differences between monkeys and sites and only gives insight in the overall sample. In other words, the resampling analyses seems to average out factors that matter, such as monkey and recording site. Furthermore, the sample is skewed toward monkey Ca which contributed about twice as much data as monkey He.

- The reader would prefer to have more insight into the variability between monkeys and between recording sites, e.g. plots with one data point per electrode. Were the effects presented here valid when the data of the monkeys was kept separate? I.e. was it replicated in the two individual monkeys. This concern should be addressed for all the analyses that are presented in the Ms.

- Please show the average visually driven response for oddball and distractor for the oddball effect (Fig. 1-3; in the format of Fig. 1a), per monkey, also in error trials.

- Related: what in Fig. 3b is “onset latency”, can we see the average response elicited by target and distractor, in correct and error trials per monkey?

- Can the authors give data per electrode? E.g. in scatter plots?

- The p-values are seriously inflated (e.g. $p < 1.e-41$ in Fig. 3c), because data points are not fully independent. A linear mixed model with factors session and monkey would be more appropriate.

3) The reader needs to have more information about the stimuli and task, before diving into the results. Many of these aspects are described only in the Methods section, i.e. only once the Ms is fully read and not all readers will do this. Please clarify early on:

- Was this an RT task? Please indicate this early on as well as the RTs of the two monkeys. What were the colors and sizes of the stimuli?
- How many cells/recording sites per monkey? Where were the RFs?
- Was the pop-out stimulus confined to RF?
- Priming of pop-out: the authors used blocks of 5-15 trials, this should be indicated at an earlier point in the Ms than in the Methods section?
- How was feature selection (Fig. 4b) quantified? Is there an index defined? How?
- Did the most selective columns all come from the same monkey? Can this analysis be repeated for individual monkeys?

4) The authors demonstrate that V4 neurons have a correlate of priming-of-popout, which also gave rise to shorter RTs. Was the V4 activity elicited by the pop-out stimulus on the primed/non-primed trials predictive of the shortened RT?

Smaller points

- Different researchers define priority and salience in different manners. To avoid confusion, it would be useful to clarify the way these concepts were defined early in the Ms.
 - Bottom of p. 2: “the feedback account of attentional capture puts forward that stimulus features are prioritized more or less independent” I don’t think that this is correct. Feedback should interact with the feedforward input, e.g. to produce a multiplicative scaling (e.g. Treue, & Martínez Trujillo, Nature 1999).
 - Fig. 1f,k I wondered why are density functions overlapping? Were these quartiles defined per session?
 - Were there systematic differences across sessions in RT? Were these predicted by V4 neurons or should they be explained by downstream areas?
 - The section starting with “Spatial evidence” for feedforward selection is a bit ambiguous because it could also refer to the spatial layout of the stimulus, perhaps “Laminar evidence” or “Layer evidence” is better?
 - Fig 2b,3d,4b: please add a scale bar that indicates electrode depth, e.g. measured from the surface of the cortex?
 - Fig. 2d indicate what is blue and cyan in the legend
- There also seem to be early effects in the superficial layers, correct?
- Fig. 3e: what is the color scale, i.e. what is the meaning of green and red?

- What is the unit on the y-axis of Fig. 4e? I.e. how should we read a.u.?
- The authors claim “an unexpectedly dominant role, dictating attentional control” for sensory cortex in priming-of-popout. These statements should be tuned down, however, because we do not know if the changes in base-line activity have a causal role or reflect a top-down effect from higher areas on V4 firing rates that do not play a causal role.

Reviewer #3 (Remarks to the Author):

Westerberg et al. investigate the presence of attentional selection in the feedforward sweep of activity in visual area V4. I have several comments that I would like the authors to consider:

(1) The selection accuracy depicted in Fig 1e rises above chance level past 100ms, well beyond V4 response latencies. This is perhaps the average of the effects shown in Fig 1g? When one looks at this result, it can be little confusing as it seemingly contradicts the claim that there is a selection signal present in the earliest volley of activity. I suggest that the author clarify the narrative to avoid the possibility of confusion.

(2) The population reliability analysis assumes independent variability among the neurons that are being considered as part of the pseudo-population. While the pseudo-population analysis can provide interesting insights, some of the results can be stronger than is really present because of this (strong) assumption of independence. The authors do have simultaneous recordings from across the layers in V4; what happens when the population reliability analyses is performed in these true populations where correlation structure is preserved?

(3) In general, the main results should be shown for each monkey

(4) Fig 1g: why is there a significant negative deflection for the slowest reaction time quartile? This pattern is similar to that in Fig 3a. How does one interpret this? Can the authors comment?

(5) Fig 1j: the caption for this panel is very confusing.

(6) Fig 2c: The power function curve does not appear to fit the data well

(7) Fig 2b caption: “Window shows 50ms ... cortical column”. I am not sure I understand what this means.

(8) In general, the authors refer to the term “synaptic current” rather loosely. While it is true that CSDs being the second spatial derivative of the LFPs represent local cortical currents, calling this measure “synaptic currents” is misleading.

(9) Fig 3e caption: “Red arrows ...”. I am not sure I see these red arrows.

(10) Lines 177-180: “While ... performed sufficient trials ...”. I might have missed this, but what was the error rate in the experiment?

Response to reviewers

Feedforward attentional selection in sensory cortex

Westerberg et al., 2023, Nature Communications

Reviewer #1

This paper describes a study into the neuronal nature of attentional capture. Using laminar probes, the authors can distinguish responses from different layers of the cortex and make inferences about the likely feedforward and feedback contributions to the observed responses. They see signatures of attentional capture in the earliest part of the response and in layer 4 and conclude that attentional capture is generated in a feedforward fashion. The question this manuscript addresses is interesting. The nature of attentional capture has been debated for decades and insight into the neuronal basis provides valuable information to this discussion. This study appears to provide such information. However, the manuscript is very dense which makes it difficult to properly evaluate the strength of the evidence that is provided for the conclusions. Selection choices in several analysis steps are not clearly explained in the main text. I think the argument could be strengthened a lot by a more explicit and extensive description of the analyses and specific conditions that are used to support the conclusions.

We thank the reviewer for their appreciation of the scientific question and recognition of the evidence this manuscript provides. We have addressed the reviewer's specific comments below which we believe addresses the concerns regarding the clarity of the work.

Some mechanistic interpretations of the results are stated as conclusions whereas they seem more like discussion points to me (see further comments for details):

Since the current work contrasts two hypotheses about the potential underlying mechanisms of attentional capture (feedforward vs feedback), I would expect these theories and their predictions for neural mechanisms to be described with a bit more detail in the introduction.

We have expanded the introduction to include more discussion regarding the neural hypotheses of these theories.

In the introduction, the authors state their results imply that the 'neural computations underlying attentional capture' occur within the earliest responses of V4 neurons. Here, I think they are speculating, as their data show a neural signature or correlate but not an actual computation. I'd suggest rephrasing these claims to state that these computations will have had to occur before or at the level of the early V4 response.

We have changed the text (including the aforementioned part of the introduction) to refer to our findings as a neural signature rather than measurement of the neuronal computation.

While I appreciate the in-line figures in this manuscript, I do think the figures are a bit too dense and small, especially figure 1. Is this due to a limit on the number of figures? If not, I think it would be nicer if Fig 1 would be split into two figures: one with the info from Fig1A-E, and one with the RT-dependent analyses explained in Fig1F-M. In addition, perhaps put panel B (task & stimuli) before A (responses split by task/stimulus) and I don't think panel C helps explaining the population reliability analysis (PRA) much. I first thought there was something different about the stimuli before realizing it's merely a visualization of the number of responses/units that are included in the analysis.

Per the reviewers request we have split the original submission Fig 1 into separate figures (now Figs. 2, 3, 4). We have also added a new figure (Fig. 1) which goes into more detail about the task and behavior prior to exploring neural data. For this revision, we have not removed what was Fig. 1c as suggested, as we have had comments from colleagues who have found that visualization helpful. Instead we include it as more of a tutorial figure to introduce the PRA. We have made it more clear in the figure legend that this does not represent a change in the stimuli, but rather an abstract representation of PRA.

Another problem with the panels in Fig 1 is the different selection criteria for the data in the panels based on RT, population size, bin of the PRA, etc. Altogether, it takes a very long time for the reader to parse figure 1 and its meaning for the result claims. This should be explained more extensively.

We believe that splitting original Fig. 1 into 3 separate figures helps the reader with parsing. In addition, we have included more text in the Results to aid in the interpretation of each of the panels per the reviewer's request.

The study reports on data from two monkeys but all results are pooled. Are the conclusions supported by results in both animals? In other words, are the same result patterns observed in individual monkeys?

We have now included a figure (Fig. 6) dedicated to demonstrating that the primary evidence is observable in each monkey.

In Figure 1f,i the difference between the colors of the two fastest response groups is difficult to see. In Fig 1i the time is relative to the sensory response time. How is this determined?

We have moved the labels in what is now Fig. 3a to boost clarity between 'Fastest' and 'Faster'. The sensory response time is the average neural response latency measured through the methods documented in the 'Feedforward sensory response window' section. Briefly, we measure the time at which each unit, on average, reaches 50% of its maximum response during the visual stimulation epoch. An average of averages is then taken as the start of the response window (found to be 58 ms following array onset). We have added that information to the Results in addition to Methods.

The MUA traces for responses at different RT classes are not shown, only the 'selection accuracy' traces of the population reliability analysis. Since this is a derived signal, it would be good to also show the average oddball vs distractor traces for these same classes (as in Fig 1a). This holds for other analyses in the paper as well. I appreciate the PRA but feel it is more of secondary analysis after showing the MUA timecourses and not the only thing to show.

We have now included the multiunit traces for that accompany the selection accuracy plots as requested by the reviewer. We also ran a Kruskal-Wallis test to show that the observed differences in the RT-quartiled target responses are significant. These can be found in the new Fig. 1 and Results.

To what extent does the relationship between the RT and population response (Fig 1m) depend on the chosen bin. Here 58-78 ms is chosen. The method section describes how this window was determined but in the main text this should also be briefly explained.

We have prepared additional panels to accompany the RT prediction analysis (Fig. 4d-e) that explore the chosen window any variations of that window in time. We have also added a description of the rationale for the chosen window to the Results section per the reviewer's suggestion.

The laminar analysis feels under-reported. It is only done on the fastest RT trials. Are similar patterns present in slower trials as well? Furthermore, the authors focus on the initial volley of activity in layer 4, but similar 'bumps' can be seen in layer 2/3 and (to a slightly lesser extent) layer 5/6 as well. These are not really discussed. What do they mean for the feedforward hypothesis?

The laminar analyses contrast fastest and slowest trials for comparison. We also included all 3 additional CSD-difference plots, however we only observe the rapid selection in the fastest condition, as initially reported. We also included discussion of the superficial and deep selection traces to the Results, as the reviewer suggested. We initially chose to focus on the early middle layer selection as that was the primary prediction of the feedforward hypothesis. Extragranular selection (as noted by the reviewer) should be expected given the canonical flow of activation through the visual cortical laminar microcircuit.

In the analysis of the error trials, responses are now not split by RT and only performed on an early time window of the response, correct? I suppose it is more difficult to do more fine-grained analysis here because of the fewer error trials, but these differences in inclusion criteria should be more clearly described and motivated.

We have addressed the lack of quartiles in the error analysis per the reviewer's request. The reviewer is indeed correct. Given the much lower occurrence of error trials, segmenting the data into quartiles yields too few trials per condition to generate interpretable findings.

In the priming analysis, can the authors comment on the stimulus response above baseline for the primed vs unprimed

oddballs? With the raised prestimulus baseline, the response increase to the stimulus seems comparable for primed and unprimed cases.

The reviewer is correct, the stimulus-evoked response change is highly similar between the unprimed and primed condition. As such, it seems as though the persistent modulation does not produce a more complex interaction with the sensory response. Seemingly, it only tonically raises the firing rate of units preferring the pertinent stimulus feature. We have noted this in the Results section.

For the section ‘Ffw selection does not require priming’, what time window is used to determine spiking responses for the Bayesian modeling and why (same goes for Fig 1)?

We used a 20 ms window from 58-78 ms following array onset. This was chosen as we believe a longer window begins to include secondary (putatively feedback-directed) responses. We chose to start the window at 58 ms as that was the average spiking response latency (as reported in the Methods section). This can be more thoroughly evaluated through the panels in new Fig. 4.

In the discussion, the authors state that their findings indicate that sensory cortex ‘dictate attentional capture’. This might be true, but cannot be conclusively concluded from the fact that signatures of attentional capture are present in the early response of neurons in feedforward layers. In addition, there is no evidence that ‘this signal is used for tonic modulation via feedback’. Any other signal in the chain from sensory code to response could also drive this feedback. The mechanism can definitely be discussed but right now it is stated as fact, which seems a step too far.

We agree. We have toned down the language accordingly.

The links to code and data currently do not yet go anywhere. I suppose this will be made public upon acceptance/publication, but would like the authors to comment on that.

We apologize for the confusion. The original link was broken and should now be fixed. As for the data, yes, we can confirm that it has been uploaded to Data Dryad and will be made public upon publication.

Reviewer #2

Westerberg et al. investigate the influence of pop-out and priming-of-popout on the activity of neurons in area V4 and how this activity relates to the reaction times of monkeys trained to make an eye movement to the pop-out stimulus. This study is a valuable addition to the literature on visual search and pop-out. However, I do have a number of concerns that need to be addressed before I can recommend acceptance of the Ms.

We thank the reviewer for their appreciation of the value of this work. We address each of their comments below.

Major points:

I would like to see a direct comparison between the feedforward response, e.g. visible in Fig. 1a and the response modulation. My impression is that the modulation comes later than the feedforward response, i.e. that seems to be visible in Fig. 1a. If this is indeed the case, i.e. the modulation occurs later than the feedforward response, then the conclusion that “[...] this relationship emerged in the earliest parts of the stimulus-driven response” (first sentences of the discussion) is not supported by the data. To address this point, the authors should directly compare the time-course of the visually driven response to the time-course of the pop-out modulation. I suspect that the visual response will precede the pop-out modulation.

We have now included the average visual responses to targets and distractors as well as the difference between the two (pop-out modulation) in Fig. 1. When not considering the ultimate behavioral RT on a given trial, the reviewer is correct: The modulation occurs after the visual response and seemingly during the time period where one might expect feedback-driven modulation. This is corroborated by our previous reports of V4 pop-out modulation which also did not consider RT as carefully (e.g., Westerberg et al., 2020, eNeuro; Westerberg et al., 2022, eLife). However, as can now be seen in the new Fig. 1, there is modulation of the target response across trials which follow the eventual RT. Moreover, this modulation begins right at the onset of the visual response. In that sense, the reviewer’s intuition is incorrect. We hope the reviewer finds the new Fig. 1 compelling.

- Same for the CSD: can the authors directly compare the time-course of the stimulus driven CSD to that caused by pop-out?

We have added the stimulus evoked CSD responses for the fastest quartile (where we observe the measurable difference) to Fig. 5 per the reviewer's request.

- The authors defined the window of the feedforward response as the 20 ms following the time at which the mean population spiking response first reaches 50% of its maximum response. That seems to imply that the earliest spikes, the ones that are really feedforward are not considered.

New Fig. 4 now includes a panel (e) which explores how well the 20 ms V4 spiking window predicts the eventual RT in a moving fashion. Here you can see that the RT can be predicted even when moving the window earlier in time (thereby accounting for the earliest spikes the reviewer mentions may be missed). Moving the window even 10 ms earlier in time, when those missed spikes are now included (so that the window is now 48-68 ms following array onset), the V4 population responses still predicts RT reliably. We believe this additional analysis satisfies the reviewers concern regarding the validity of the window chosen.

- The authors may want to relate their results to those of Poort et al. (Neuron, 2012) demonstrating that the V4 visual response precedes a salience response driven by figure-ground segregation (which is related to the pop-out signal described here) by about 15ms.

We now include a discussion of similarities between pop-out search and figure-ground segmentation in the discussion. As for the differences in the latency, that could be related to the overall saliency of the stimuli – a red vs. green color difference could be argued as more salient than a stimulus of one orientation on a background of another. Another possibility is that the level at which we are able to detect that modulation is entirely dependent on having a number of different RT bins that we organize the data into. This is different than the latencies which are measured in Poort et al., 2012, Neuron. It is also worth noting that we are not claiming that the selection we observe is due purely to a salience gradient, only that the target (oddball) response can be robustly related to the ultimate RT in the task. We have now made this clear in the discussion as well.

- If the effect is indeed visible in the feedforward response, i.e. in the very first spikes, the reader wonders where the information, that requires a comparison between the stimulus in the RF and the other stimuli can come from. In other words, what is the signal path from the stimuli outside the RF to the V4 neurons that is short enough to influence the first V4 spikes, which are presumably driven from the V4-RF itself?

We now make clearer that we do not claim that V4 is the first place that this occurs. It is entirely possible (likely even given what is known of figure-ground segregation) that V1 already is contributing to this selection process. In the way that horizontal connections seem to contribute to figure-ground segregation, the same could be hypothesized for the attentional capture documented here. We have included this consideration in the Discussion section as a putative mechanism.

It remains unclear whether the effects were replicated in both monkeys. The resampling analysis seems inappropriate from a statistic point of view. The reason is that the resampling approach considered trials to be statistically independent, irrespective of whether they come from one or the other monkey (and recording session – although that presumably plays a smaller role here). Sampling from all the trials and recording sites will obscure differences between monkeys and sites and only gives insight in the overall sample. In other words, the resampling analyses seems to average out factors that matter, such as monkey and recording site. Furthermore, the sample is skewed toward monkey Ca which contributed about twice as much data as monkey He.

Unfortunately, we are unable to evaluate the data through these methods with completely independent samples. To do so, we would need simultaneous recordings across V4 locations with receptive fields corresponding to each of the array-item positions. However, we believe three newly included analyses may help with alleviating the reviewer's concerns. (1) We now show that there is an observable (and significant) difference in the average multiunit responses to the target stimulus (Fig. 1). These methods are more like traditional reports of spike-rate differences which have been the standard of the field for decades. More directly addressing the comment, (2) We now supply the main findings of the population

reliability analysis at the individual monkey level (Fig. 6), thereby demonstrating that this was not an effect driven by only one monkey but is instead observed in both. And (3), we have included a population reliability analysis where we preserve the independence of the sample representing each item for a selection simulation (Fig. 10). The latter shows that the differences in selection profiles are apparent when population response are restricted to a simultaneously sampled set of multiunits.

- The reader would prefer to have more insight into the variability between monkeys and between recording sites, e.g. plots with one data point per electrode. Were the effects presented here valid when the data of the monkeys was kept separate? I.e. was it replicated in the two individual monkeys. This concern should be addressed for all the analyses that are presented in the Ms.

We have now included a new figure (Fig. 6) which demonstrates the main effects in each monkey for the analyses for which we found sufficient statistical power.

- Please show the average visually driven response for oddball and distractor for the oddball effect (Fig. 1-3; in the format of Fig. 1a), per monkey, also in error trials. Related: what in Fig. 3b is “onset latency”, can we see the average response elicited by target and distractor, in correct and error trials per monkey?

Visually-driven responses for correct trials in their respective quartiles are now shown as an average of all sessions/monkeys in Fig. 1 and for each monkey in Fig. 6. Error responses for both animals and each animal are now shown in Fig. 7.

- Can the authors give data per electrode? E.g. in scatter plots?

We now provide per-electrode data in a histogram detailing the slope of the change in spike rate as a function of RT rank.

- The p-values are seriously inflated (e.g. $p < 1.e-41$ in Fig. 3c), because data points are not fully independent. A linear mixed model with factors session and monkey would be more appropriate.

We agree with reviewer. Unfortunately, we are not able to perform the analysis the reviewer suggested since the data are not generated from the monkey or session level, but are derived from the PRA. We are also not able to determine significance in the same way as we do for the other data generated via PRA because those rely on the variability in the baseline data to determine significance. In this revision, we have chosen to eliminate the test statistic associated with the result and instead note some characteristics of that finding so that the reader can deem whether or not it is compelling.

3) The reader needs to have more information about the stimuli and task, before diving into the results. Many of these aspects are described only in the Methods section, i.e. only once the Ms is fully read and not all readers will do this.

We agree that additional experimental details would be useful in advance of diving into the results. We have modified the results section to reflect pertinent information per the reviewer’s recommendations detailed below.

- Was this an RT task? Please indicate this early on as well as the RTs of the two monkeys. What were the colors and sizes of the stimuli?

This information has been added to the first section of the Results per the reviewer’s request.

- How many cells/recording sites per monkey? Where were the RFs?

15 sites were used from each recording session (n=29 sessions, n=2 monkeys) totally 435 multiunit sites (n=300 monkey Ca, n=135 monkey He). RFs were mapped for each session. In monkey Ca, RFs were found 2-10 dva eccentricity in the lower righthand visual quadrant. In monkey He, RFs were found 2-10 dva eccentricity in the lower lefthand visual quadrant. These correspond well to the expected location of RFs given the position of each monkey’s recording chamber (left-side chamber, monkey Ca; right-side chamber, monkey He) over the prelunate gyrus. Sessions were only included if we were able to map RFs and the RFs were preserved along cortical depth. RFs were more thoroughly reported in two previous manuscripts (Westerberg et al., 2021, PNAS; Westerberg et al., 2022, eLife).

- Was the pop-out stimulus confined to RF?

Yes, the RF was mapped on a session-by-session basis and the stimulus was restricted to the RF. The stimulus was made to be smaller than the average size of a V4 receptive field at a given eccentricity. We did not employ a size tuning paradigm prior to a given recording session due to time constraints, but we did use a stimulus that scaled with eccentricity by 0.3 dva per 1 dva eccentricity whereas the putative V4 RF scales 0.84 dva per 1 dva eccentricity (Freeman and Simoncelli, 2011, Nat Neurosci).

- Priming of pop-out: the authors used blocks of 5-15 trials, this should be indicated at an earlier point in the Ms than in the Methods section?

This point is now included early in the Results section per the reviewer's request.

- How was feature selection (Fig. 4b) quantified? Is there an index defined? How?

Feature selection was calculated by performing a variation of the population reliability analysis at the individual unit level. In essence, the magnitude of red vs. green responses are compared in the distractor condition for each unit to determine which color elicits a greater response. The exact details are outlined in the methods. Notably, these criteria matched the feature selectivity reported in a previous manuscript using the same data but determined with a different feature selectivity metric (Westerberg et al., 2021, PNAS). The index in this manuscript (reported as a %) refers to the percent of time a sample of one color for a given unit elicits a greater response than the other color. We have added text to the Results to elaborate this approach.

- Did the most selective columns all come from the same monkey? Can this analysis be repeated for individual monkeys?

No, of the 12 included columns, 9 came from monkey Ca (75%) and 3 from monkey He (25%). This is close to the relative proportions of recording sessions performed in each animal (monkey Ca, 69%; He, 31%). However, we are unable to perform this analysis at the individual animal level as 3 sessions in 1 animal are too few to perform the statistical test we did at the entire population level. Unfortunately, it is also impossible to acquire more data from this animal. We hope the reviewer finds the analyses we provide at the individual animal level convincing in that the differences between animals do not appear to affect the overall conclusions of this manuscript.

4) The authors demonstrate that V4 neurons have a correlate of priming-of-popout, which also gave rise to shorter RTs. Was the V4 activity elicited by the pop-out stimulus on the primed/non-primed trials predictive of the shortened RT?

Yes, in fact we have an entire paper (Westerberg et al., 2020, eNeuro) relating the responses of V4 units to the ultimate RT in priming of pop-out (PoP). We now include a more explicit statement in the priming section directing the reader to that manuscript should a more thorough evaluation of V4 to PoP be of interest.

Smaller points:

- Different researchers define priority and salience in different manners. To avoid confusion, it would be useful to clarify the way these concepts were defined early in the Ms.

A discussion of salience vs. priority is now included and expanded on in the first paragraph of the Results section per the reviewer's request.

- Bottom of p. 2: "the feedback account of attentional capture puts forward that stimulus features are prioritized more or less independent" I don't think that this is correct. Feedback should interact with the feedforward input, e.g. to produce a multiplicative scaling (e.g. Treue, & Martínez Trujillo, Nature 1999).

We agree with the reviewer. We have altered the text to not espouse that viewpoint.

- Fig. 1f,k I wondered why are density functions overlapping? Were these quartiles defined per session?

The reviewer is correct, the quartiles were defined at the session level. This has been made clearer in the new Fig. 1 as well as the associated text in the Results section.

- Were there systematic differences across sessions in RT? Were these predicted by V4 neurons or should they be explained by downstream areas?

We observed modest differences in the RTs from session-to-session. The standard deviation of mean RTs for each animal were found to be 22 ms in monkey Ca (8.7% change) and 10 ms in monkey He (4.5% change). However, in attempting to investigate this further we first found the difference between the average sensory, feedforward response (58-78 ms following array onset) for each unit to the target for the fastest and slowest quartiles (as a putative index for the range of target response values on a given session for a given unit). We then correlated that with the average RT for the session on which that unit was recorded. We found a Pearson R of 0.0025 with a p-value of 0.96. From this pilot analysis, the variability in target responses on a session-by-session basis are not capable of explaining the modest differences in session-by-session RTs. This could perhaps be taken as evidence that RT variability day-to-day might be explained by higher order differences rather than lower order differences. However, this is rather speculative.

- The section starting with “Spatial evidence” for feedforward selection is a bit ambiguous because it could also refer to the spatial layout of the stimulus, perhaps “Laminar evidence” or “Layer evidence” is better?

We have changed the section label as suggested.

- Fig 2b,3d,4b: please add a scale bar that indicates electrode depth, e.g. measured from the surface of the cortex?

We have included the scale bars as requested; however, we indicate depth relative to L4/5 boundary instead of top of cortex as that is what we used to align the data across sessions.

- Fig. 2d indicate what is blue and cyan in the legend

We have added the meaning of the colors to the legend as well as the figure itself.

There also seem to be early effects in the superficial layers, correct?

Yes, this observation is now added to the Results and expanded on. We focus on the middle layers as the prediction of the feedforward hypothesis is predicated on early dissociable activation in the feedforward layers.

- Fig. 3e: what is the color scale, i.e. what is the meaning of green and red?

The interpretation of the colors with reference examples are now included in the figure legend.

- What is the unit on the y-axis of Fig. 4e? I.e. how should we read a.u.?

After computing the multiunit envelope as described in the methods, the activity was normalized at the trial level using a z score method where the standard deviation was taken in the 100 ms window before stimulus presentation. A.U. therefore indicates the magnitude of the activity with respect to the variability in the baseline period. This has been noted in the methods.

- The authors claim “an unexpectedly dominant role, dictating attentional control” for sensory cortex in priming-of-popout. These statements should be tuned down, however, because we do not know if the changes in base-line activity have a causal role or reflect a top-down effect from higher areas on V4 firing rates that do not play a causal role.

We have toned down the language as requested.

Reviewer #3

Westerberg et al. investigate the presence of attentional selection in the feedforward sweep of activity in visual area V4. I have several comments that I would like the authors to consider:

We thank the reviewer for their careful consideration of the manuscript. We respond to their comments below:

(1) The selection accuracy depicted in Fig 1e rises above chance level past 100ms, well beyond V4 response latencies. This is perhaps the average of the effects shown in Fig 1g? When one looks at this result, it can be little confusing as it seemingly contradicts the claim that there is a selection signal present in the earliest volley of activity. I suggest that the author clarify the narrative to avoid the possibility of confusion.

We have now separated those results into different figures to not promote that direct comparison. We have also noted what the reviewer mentioned regarding the average and expanded on how the power of this analysis lies in evaluating V4 response as a function of RT.

(2) The population reliability analysis assumes independent variability among the neurons that are being considered as part of the pseudo-population. While the pseudo-population analysis can provide interesting insights, some of the results can be stronger than is really present because of this (strong) assumption of independence. The authors do have simultaneous recordings from across the layers in V4; what happens when the population reliability analyses is performed in these true populations where correlation structure is preserved?

We hope that the new Fig. 10 satisfies the reviewer's requests. We have preserved the independence of the samples contributing to each population response. The differences in selection profiles can still be observed with this restriction to the analysis.

(3) In general, the main results should be shown for each monkey

We have now included a new Figure (Fig. 6) which demonstrates the main effects in each monkey.

(4) Fig 1g: why is there a significant negative deflection for the slowest reaction time quartile? This pattern is similar to that in Fig 3a. How does one interpret this? Can the authors comment?

A very valuable observation! This is because one (or more) of the population responses to the distractor stimulus was greater than that of the target. Speculatively, this could mean that a distractor is errantly selected initially. This errant selection then could be the reason for the delayed response time. We have added this speculation to the results as an interesting observation that will require further investigation to unravel.

(5) Fig 1j: the caption for this panel is very confusing.

We have clarified the text for that panel.

(6) Fig 2c: The power function curve does not appear to fit the data well

The fit for CSD is indeed poorer than that of the spiking activity. Speculatively, this could be a result of the lower signal-to-noise for CSD as compared to spiking. We have noted this in the text.

(7) Fig 2b caption: "Window shows 50ms ... cortical column". I am not sure I understand what this means.

We have changed the language for clarity. It simply meant that the panel focused in time on the 50 ms window around the time of feedforward activation from the stimulus-driven visual response.

(8) In general, the authors refer to the term "synaptic current" rather loosely. While it is true that CSDs being the second spatial derivative of the LFPs represent local cortical currents, calling this measure "synaptic currents" is misleading.

We have changed the language throughout to *putative* synaptic currents as to not overstate or misrepresent the precision of the CSD estimation method.

(9) Fig 3e caption: "Red arrows ...". I am not sure I see these red arrows.

Missing red arrows have been fixed.

(10) Lines 177-180: “While ... performed sufficient trials ...”. I might have missed this, but what was the error rate in the experiment?

Error rate is now included in the new Figure 1.

REVIEWERS' COMMENTS

Reviewer #3 (Remarks to the Author):

I am satisfied with the authors' responses and the revised manuscript.

One minor point: please add time axis labels to Fig 1d

Reviewer #4 (Remarks to the Author):

To the authors and editor:

I have been asked to assess if the authors have satisfied the concerns of the reviewers and responded accordingly.

I am sorry it took me a while to go back through the review process to fully understand not only the manuscript but also the concerns raised.

In my opinion, the authors have done a great job integrating the reviewers questions and have satisfied all remaining concerns.

Response to reviewers (rev. accepted in principle)

Feedforward attentional selection in sensory cortex

Westerberg et al., 2023, Nature Communications

Reviewer #3

I am satisfied with the authors' responses and the revised manuscript. One minor point: please add time axis labels to Fig 1d.

We thank the reviewer for reviewing the revised manuscript. We are glad to hear that the revisions were satisfactory. We have added the time axis label to Fig. 1d as requested.

Reviewer #4

I have been asked to assess if the authors have satisfied the concerns of the reviewers and responded accordingly. I am sorry it took me a while to go back through the review process to fully understand not only the manuscript but also the concerns raised. In my opinion, the authors have done a great job integrating the reviewers questions and have satisfied all remaining concerns.

We thank the reviewer for taking the time to review the manuscript. We are happy to hear the reviewer is satisfied with the manuscript and the revisions to the previous version.